# Potent Chlorambucil-Platinum(IV) Prodrugs

**DOI:** 10.3390/ijms231810471

**Published:** 2022-09-09

**Authors:** Angelico D. Aputen, Maria George Elias, Jayne Gilbert, Jennette A. Sakoff, Christopher P. Gordon, Kieran F. Scott, Janice R. Aldrich-Wright

**Affiliations:** 1School of Science, Western Sydney University, Locked Bag 1797, Penrith South DC, Sydney, NSW 2751, Australia; 2Ingham Institute, Liverpool, NSW 2170, Australia; 3Calvary Mater Hospital, Waratah, NSW 2298, Australia

**Keywords:** PHEN*SS*, 5ME*SS*, 56ME*SS*, DNA, chlorambucil, platinum(II), platinum(IV), cytotoxicity, ROS

## Abstract

The DNA-alkylating derivative chlorambucil was coordinated in the axial position to atypical cytotoxic, heterocyclic, and non-DNA coordinating platinum(IV) complexes of type, **[Pt^IV^(H_L_)(A_L_)(OH)_2_](NO_3_)_2_** (where H_L_ is 1,10-phenanthroline, 5-methyl-1,10-phenanthroline or 5,6-dimethyl-1,10-phenanthroline, A_L_ is 1*S*,2*S*-diaminocyclohexane). The resultant platinum(IV)-chlorambucil prodrugs, **PCLB**, **5CLB**, and **56CLB**, were characterized using high-performance liquid chromatography, nuclear magnetic resonance, ultraviolet-visible, circular dichroism spectroscopy, and electrospray ionization mass spectrometry. The prodrugs displayed remarkable antitumor potential across multiple human cancer cell lines compared to chlorambucil, cisplatin, oxaliplatin, and carboplatin, as well as their platinum(II) precursors, **PHEN*SS***, **5ME*SS***, and **56ME*SS***. Notably, **56CLB** was exceptionally potent in HT29 colon, Du145 prostate, MCF10A breast, MIA pancreas, H460 lung, A2780, and ADDP ovarian cell lines, with GI_50_ values ranging between 2.7 and 21 nM. Moreover, significant production of reactive oxygen species was detected in HT29 cells after treatment with **PCLB**, **5CLB**, and **56CLB** up to 72 h compared to chlorambucil and the platinum(II) and (IV) precursors.

## 1. Introduction

Chemotherapy is the workhorse of oncology and the most effective type of cancer treatment. This effectiveness originates from a conventional class of platinum(II) anticancer agents, which includes *cis*-diamminedichloroplatinum(II) (cisplatin) and its derivatives, *trans*-*L-*(1*R*,2*R*-diaminocyclohexane) oxalatoplatinum(II) (oxaliplatin) and *cis*-diammine(1,1-cyclobutanedicarboxylato) platinum(II) (carboplatin) (Figure 1). These drugs are highly efficient in treating genitourinary cancers such as ovarian and testicular cancer, as well as small-lung, head, and neck cancers due to their DNA-binding mechanism of action [1,2,3]. Four other platinum-based drugs are also available for chemotherapeutic use, but their regulatory approval is exclusive to selected countries: nedaplatin and miriplatin (Japan), lobaplatin (China), and heptaplatin (South Korea), respectively (Figure 1) [4]. In addition to platinum, many other metal coordination complexes have been discovered and gained therapeutic interest because of their distinctive architectures and the unique properties that result [4,5]. This productive area of research is inspired by the desire to discover cancer’s magic bullet. While there is plenty of exploration of other metal coordination complexes, platinum-coordination complexes (i.e., platinum(II) drugs) continue to be the mainstay of chemotherapy.

Much of the success and popularity of platinum(II) drugs stem from the unforeseen scientific discovery of Rosenberg, VanCamp, and Krigas, who investigated the effects of electricity on bacterial cell growth [6]. Initially, it was assumed that electricity had a significant effect on cell division but later it was discovered to be the effect of a platinum(II)-based complex formed from the oxidation of the electrode. This complex was identified as cisplatin and after a cascade of preclinical studies and clinical trials, it became the first metal-based drug to be approved in 1978 and 1979, in the U.S. and U.K., respectively as a chemotherapeutic for the treatment of genitourinary cancers [7]. Following the clinical implementation of cisplatin and years of combined efforts by researchers worldwide, with the search for additional anticancer drugs have resulted in the development of oxaliplatin and carboplatin [8,9].

While platinum(II) drugs are extremely robust anticancer agents, dose-limiting side effects constraint their use [10,11,12]. Fatigue, nausea, vomiting, constipation, hair loss, kidney failure, liver failure and nerve damage are the most common side effects observed in cancer patients during chemotherapy [13,14,15,16,17,18]. Cancer patients are also at risk of developing secondary complications such as nerve hypersensitivity, vascular issues and immunosuppression due to the aforementioned side effects [16,19,20,21]. The toxicity of platinum(II) drugs results from poor selectivity, which means that such drugs cannot discriminate between normal healthy cells and cancer cells [22,23]. As a result, normal healthy cells also experience genomic instability, which is how side effects and secondary complications arise. Consequently, these drugs are likely to react with ribonucleic acid, thiols, glutathione, phospholipids, and other biomolecules in the body that contain nitrogen, sulfur and oxygen atoms, in addition to cellular deoxyribonucleic acid (DNA) (Figure 2) [22,23,24,25].

A promising strategy employed to diminish the limitations associated with platinum(II) drugs is to develop new platinum(IV) complexes. Accordingly, a series of comprehensible reviews have highlighted the likelihood of platinum(IV) complexes as the future of platinum-based drugs in chemotherapy [26,27,28,29,30,31]. Platinum(IV) complexes have a six-coordinate octahedral geometry where the two axial positions allow the coordination of additional ligands in contrast with square planar geometry of platinum(II) complexes (Figure 3) [32]. Generally, conjugation of axial positions is sequentially achieved by reacting platinum(II) complexes with oxidizing agents such as hydrogen peroxide or chlorine [27]. The consequent hydroxyl/chloride functionalities serve as versatile, functional handles to append either bioactive or non-bioactive scaffolds. To activate a platinum(IV) complex, it must undergo reduction, which occurs once the complex is in the cell, where the axial ligands are released in conjunction with the formation of the platinum(II) precursor (Figure 3) [26,33]. Activation of the platinum(IV) complexes within the cell is useful because they are more stable in the bloodstream due to their low spin d^6^ electronic configuration [27]. This unique advantage can mitigate unwanted interactions with off-target biomolecules in the body, and accordingly limits side effects or other clinical complications; thus offering improved drug selectivity [26].

A major advantage of platinum(IV) design is the possibility for oral administration, which was previously demonstrated by *bis*-(acetate)-ammine dichloro-(cyclohexylamine) platinum(IV) (satraplatin). Satraplatin was the first orally available platinum-based chemotherapeutic prodrug that entered phase III clinical trials but later failed because an overall increase in survival benefit was not demonstrated [34,35,36]. The availability of affordable, effective, and orally administered anticancer drugs would improve the cancer treatment experience for patients. Moreover, it would be more convenient for cancer patients, as hospitalization would not be necessary.

Numerous examples of multi-action platinum(IV) prodrugs incorporating bioactive axial ligands that target or inhibit polypeptides and enzymes have been reported in the literature [28,30,37]. However, research into the conjugation of a DNA-alkylating drug such as 4-[*bis*(2-chloroethyl)amino] phenylbutyric acid (brand name: Leukeran, or chlorambucil (CLB)) as an axial ligand to platinum is limited. This provoked our interest to explore platinum(IV) prodrugs, using the cores of heterocyclic, non-DNA coordinating platinum(II) complexes such as [Pt^II^(1,10-phenanthroline)(1*S,*2*S*-diaminocyclohexane)]Cl_2_ (**PHEN*SS***), [Pt^II^(5-methyl-1,10-phenanthroline)(1*S,*2*S*-diaminocyclohexane)]Cl_2_ (**5ME*SS***) and [Pt^II^(5,6-dimethyl-1,10-phenanthroline)(1*S,*2*S*-diaminocyclohexane)]Cl_2_ (**56ME*SS***), with CLB (Figure 4). Our research group has previously reported these atypical class of platinum(II) complexes [38,39,40,41]. They can be represented as **[Pt^II^(H_L_)(A_L_)]^2+^** where H_L_ is the heterocyclic ligand (i.e., 1,10-phenanthroline, 5-methyl-1,10-phenanthroline or 5,6-dimethyl-1,10-phenanthroline) and A_L_ is the ancillary ligand (i.e., 1*S,*2*S*-diaminocyclohexane (*S*,*S*-DACH)) [38,39,40,41]. Complexes of this type, especially **56ME*SS***, are of interest due to their superior biological activity compared to cisplatin, oxaliplatin and carboplatin in several cancer cell lines [42,43]. Although the mechanism of action of **56ME*SS*** is not fully elucidated, it is purported to interfere with the mitochondria by reducing the mitochondrial membrane potential and modify the architecture of cytoskeletal networks and nuclear DNA [43,44]. We hypothesize that **PHEN*SS*** and **5ME*SS*** act the same way as **56ME*SS***, due to their structural similarities.

CLB belongs to an earlier generation of anticancer agents known as nitrogen mustards. Similar to other within the class, CLB elicits potency through binding with nucleotides of the DNA, such as guanine and adenine, at the *N*(7) and *N*(3) positions, generating DNA crosslinks [45,46]. CLB is approved for chemotherapeutic use against Hodgkin’s lymphoma, ovarian cancer, chronic lymphocytic leukemia, as well as certain types of trophoblastic neoplasms [45]. Unfortunately, the potency of CLB as an alkylating agent is impaired by its low bioavailability and poor selectivity toward cancer cells, often resulting in near intolerable side effects [47,48,49]. A recent study reported that upon administration of CLB, the drug is instantly detoxified by human glutathione transferase Pi, an enzyme found over-expressed in cancer tissues [50]. When CLB is conjugated to glutathione, alkylating proficiency is reduced, making the drug no longer effective, especially against cancer cells with a developed resistance. A potential strategy to resolve this quandary is to use a platinum(IV) scaffold as a delivery vehicle for CLB. Recent studies have reported the synthesis of cisplatin/oxaliplatin-CLB prodrugs to create dual mode-DNA-binding complexes that improved the bioavailability of CLB and most importantly, overcome chemoresistance. Ma et al. synthesized two platinum(IV) derivatives of cisplatin bearing CLB to assess whether coordinating CLB improved the cellular uptake and DNA-binding of the resulting prodrugs [51]. All the platinum(IV) derivatives bearing CLB from the study displayed superior cytotoxicity, most prominently toward cisplatin-resistant cancer cell lines, which was also confirmed by Qin et al. in a separate study [52]. It is worth mentioning that the CLB released from the platinum(IV) derivatives, induced an increased level of DNA damage by blocking the double-strand break repair process [51]. A positive aspect of this design is that these platinum(IV) derivatives containing CLB in the axial position exhibited longer retention times with almost no untargeted toxicities in vivo [51]. Furthermore, Montagner et al. undertook a similar investigation with findings that supported the claims of Ma et al. and Qin et al., and established that the CLB effectively tunes the lipophilicity of the platinum(IV) prodrug [53]. Lipophilicity plays an important role in terms of fighting cisplatin-resistant cancer cells, as it improves cellular uptake properties and it is hypothesized to facilitate passive diffusion through the cell membranes [26]. Pathak et al. corroborated the effectiveness of conjugating CLB to cisplatin in fighting chemoresistance by encapsulating the prodrug (cisplatin-CLB) in the hydrophobic core of polymeric nanoparticles designed to target the mitochondria [54]. Remarkably, these mitochondria-targeted nanoparticles allowed the efficacious delivery of cisplatin-CLB to the mitochondria, thereby demonstrating greater cytotoxicity compared to the prodrug itself. These studies validate that a platinum(IV) prodrug composed of two DNA-damaging agents (i.e., CLB and cisplatin) can overcome chemoresistance. Most importantly, these studies contribute to our understanding that the clinical limitations of CLB as a single drug can be compensated for by coordinating it to a platinum(IV) complex to produce a prodrug with improved cancer-killing properties.

Herein, we report the synthesis, characterization, and biological investigations of three non-symmetrical platinum(IV) prodrugs incorporating the **[Pt^II^(H_L_)(A_L_)]^2+^** scaffold, where the two axial positions are occupied by CLB and a hydroxide group (OH) (Figure 4). The chemical structures of the synthesized platinum(IV)-CLB prodrugs, [Pt^IV^(1,10-phenanthroline)(*S*,*S*-DACH)(CLB)(OH)](NO_3_)_2_ (**PCLB**), [Pt^IV^(5-methyl-1,10-phenanthroline)(*S*,*S*-DACH)(CLB)(OH)](NO_3_)_2_ (**5CLB**) and [Pt^IV^(5,6-dimethyl-1,10-phenanthroline)(*S*,*S*-DACH)(CLB)(OH)](NO_3_)_2_ (**56CLB**) are illustrated in Figure 4. The in vitro cytotoxicity of the prodrugs and their core scaffolds were evaluated on a panel of cell lines. The potential cellular reactive oxygen species (ROS) production of these prodrugs and their platinum(II) and (IV) scaffolds, together with CLB, cisplatin, carboplatin, and oxaliplatin, was also assessed on the cisplatin-resistant colorectal cancer line: HT29.

## 2. Results and Discussion

### 2.1. Synthesis and Characterization

The complexes presented in this work are non-symmetrical platinum(IV) derivatives of **[Pt^II^(H_L_)(A_L_)]^2+^** with CLB coordinated as a bioactive axial ligand. The synthesis pathway (Figure 5) employed to create the prodrugs was devised through the optimization of previously described techniques [43,51,52,53,54,55,56,57]. Firstly, **[Pt^II^(H_L_)(A_L_)]^2+^** was synthesized and characterized by NMR (Appendix A), with results consistent with the literature data [57,58,59,60,61,62]. Secondly, **[Pt^II^(H_L_)(A_L_)]^2+^** was oxidized to afford **[Pt^IV^(H_L_)(A_L_)(OH)_2_]^2+^** [57,58,59,60,61]. **[Pt^IV^(H_L_)(A_L_)(OH)_2_]^2+^** was also characterized by NMR (Appendix A), with results consistent with the literature data [57,58,59,60,61]. The symmetrical anhydride of CLB was derived from a modified literature protocol [54]. Although the CLB anhydride could not be fully isolated and characterized, HPLC was still used to check its presence (Appendix A). The CLB anhydride was used without purification and reacted with **[Pt^IV^(H_L_)(A_L_)(OH)_2_]^2+^** to afford **[Pt^IV^(H_L_)(A_L_)(CLB)(OH)]^2+^** (Figure 5). The desired products were purified by a flash chromatography system and thoroughly characterized using HPLC (Appendix A), ^1^H-NMR (Appendix A), 2D-COSY (Appendix A), ^1^H-^195^Pt-HMQC (Appendix A), UV (Appendix A), CD (Appendix A) and ESI-MS (Appendix A). **[Pt^IV^(H_L_)(A_L_)(CLB)(OH)]^2+^** is soluble in 10 mM PBS (~7.4 pH), 0.9% NaCl and d.i.H_2_O and the solubility was determined to be 5 mg/mL at room temperature in each.

#### 2.1.1. H-NMR and ^1^H-^195^Pt-HMQC Spectral Assignment

A summary of the ^1^H-NMR and ^1^H-^195^Pt-HMQC data of **PCLB**, **5CLB**, and **56CLB** is presented in Table 1. Because some of the structural features of the complexes are symmetrical, some protons are chemically equivalent. No amine proton resonances were observed due to proton exchange with D_2_O. Protons were assigned according to the proton labeling system provided in Figure 6.

Using **PCLB** as an example, its ^1^H-NMR spectrum shows slightly shifted downfield multiplicity (Figure 7), which is attributed to the coordination of CLB to the platinum. This is also due to the electrophilic nature of CLB as it contains electronegative atoms such as nitrogen and chlorine as well as an aromatic ring (π bonding), hence causing protons to be deshielded. As shown in Figure 7, the aromatic protons originating from the heterocyclic ligand, phenanthroline, specifically H2 and H9 resonated as two adjacent doublets at 9.30 and 9.27 ppm, respectively. H2 and H9 protons were expected to resonate furthest downfield because of the deshielding effect caused by the electronegativity of the nitrogen atoms in the phenanthroline ring, which also exhibits π bonding. H4 and H7 protons also resonated as doublet of doublets at 8.99 ppm, with calculated *J*-coupling constants of 8.3 and 3.3 Hz. H3 and H8 protons appeared as a multiplet at 8.24 ppm, slightly overlapping with the singlet at 8.20 ppm derived from H5 and H6 protons. Normally, the resonances observed for the phenanthroline ligand of **PCLB** are different to what is observed for its platinum(II) and (IV) precursor complexes, **PHEN*SS*** and **PHEN*SS*(IV)(OH)_2_** because they appear as doublets and, more importantly, no overlapping is involved between H3 and H8 and H5 and H6 protons (Figure 8) [57,58,59,60,63]. Clearly, the differences in multiplicity are a consequence of axial coordination of the CLB ligand.

Furthermore, the aromatic protons of CLB represented by a, b, c, and d appeared as a quartet at 6.37 ppm, with a *J*-coupling constant of 8.8 Hz. The sharp resonance at 3.51 ppm with an integration of 8 protons was assigned to the methylene protons of CLB (the mustard end), which are represented by e, f, g, and h. Moreover, the resonances within the aliphatic region (1–3 ppm) originating from the ancillary ligand, DACH are consistent with what is reported in the literature [56,57,58,59,60,63], although the resonances are slightly more upfield due to the coordination of CLB. The β and γ protons of CLB (methylene protons near the aromatic ring) overlapped with the signals of DACH protons as shown in Figure 7 and Figure 8. Clearly, γ was expected to shift further downfield than β because of their proximity to the aromatic ring of CLB.

To further confirm that the CLB ligand has been successfully coordinated to the platinum and only occupied one axial position, ^1^H-^195^Pt-HMQC experiments were performed in the regions of −2800 and 400 ppm. Generally, platinum(II) resonates at −2800 ppm while platinum(IV) resonates at 400 ppm as previously reported [56,57,58,59,60,63]. According to the ^1^H-^195^Pt-HMQC spectrum of **PCLB** in the 400 ppm region (Figure 9), two peaks were recorded at 542 ppm highlighting the correlation of the H2, H9, H3, and H8 protons to the platinum.

The ^1^H-NMR spectrum of **5CLB** (Appendix A) displayed a few variations compared to the results obtained for **PCLB** and **56CLB** (Appendix A), particularly the resonances along the aromatic region. Both H2 and H9 protons resonated as a quartet of doublets at 9.26 ppm. Additionally, H4 and H7 protons resonated as two separate signals or doublets whereas in the ^1^H-NMR spectrum of **PCLB** and **56CLB** (Appendix A), these signals were combined. H3 and H8 protons of **5CLB** also resonated as two individual multiplets at 8.26 and 8.19 ppm (Table 1 and Appendix A), respectively. As for the ^1^H-NMR spectrum of **56CLB** (Appendix A), slight variations were also noted. For example, greater distance between the resonances derived from the CLB aromatic protons, a and b and c and d, were also observed while in the ^1^H-NMR spectrum of **PCLB** (Figure 7 and Figure 8) and **5CLB** (Appendix A), where these resonances were adjacent. Furthermore, two of the methylene protons of CLB represented by γ and β protons overlapped with the signals of the least deshielded DACH protons (H4′ and H5′ ax.), exhibiting a multiplet at 1.20 ppm (Table 1 and Appendix A). Overall, these variations in multiplicity are mainly attributed to the structure of the complexes. **5CLB** and **56CLB** both contain methyl substituents in their phenanthroline systems, while **PCLB** has none. From this, it can be concluded that methyl groups in the phenanthroline system can influence variations in multiplicity. Nevertheless, the ^1^H-^195^Pt-HMQC results obtained for **5CLB** (Appendix A) and **56CLB** (Appendix A) also confirmed the successful coordination of CLB ligand to their platinum cores.

#### 2.1.2. UV and CD Studies

The electronic transitions observed in the UV measurements were comparable to published examples, especially those of platinum(II) and (IV) precursor complexes [57,58,59,60,61,64]. The UV spectra of the platinum(IV)-CLB complexes (Appendix A) were obtained by titrations for each complex, in triplicate to attain consistency and accuracy. Molar extinction coefficients were also calculated based on a generated plot of absorbance against concentration using the Beer–Lambert law equation. Standard errors were also calculated. In conjunction with UV, CD measurement were recorded to confirm the retention of chirality of the ancillary ligand, *S*,*S*-DACH. The obtained CD spectra (Appendix A) are also comparable to published examples [57,58,59,60,64]. All characteristic peaks in the UV and CD spectra of the complexes are summarized in Table 2.

The results obtained from the UV measurements demonstrated both π–π* transitions and metal-to-ligand charge transfer interactions (Appendix A). Generally, π–π* transitions are attributed to conjugated ring systems such as the heterocyclic ligands (i.e., phenanthrolines) present in the structure of the studied platinum(IV)-CLB complexes. From the UV spectra of **PCLB** (Appendix A), a strong absorption band was recorded at 203 nm, followed by another prominent band at 279 nm and also, a weaker shoulder featured at approximately 306 nm. In comparison, the UV spectra of **5CLB** (Appendix A) and **56CLB** (Appendix A) also showed strong absorption at 204 nm, followed by weaker absorption bands at 284 and 291 nm, respectively. For comparison purposes, Figure 10 illustrates the absorption variations exhibited by **PCLB**, **5CLB**, and **56CLB**. It is evident that strong absorption bands peaked around 200 nm for the complexes, followed by weaker absorption bands between 260 and 300 nm. Specifically, **56CLB** exhibited a distinct absorption band (hyperchromic effect) between 235 and 250 nm, opposite to what was observed for **PCLB** and **5CLB** wherein decreased absorption (hypochromic effect) was demonstrated (Figure 10). Accordingly, there is also evidence of a bathochromic effect or ‘red shift’ as reflected by the movement of the absorption bands recorded for each complex between 270 and 300 nm (Figure 10). This phenomenon is likely a result of the absence or presence of methyl group on the phenanthroline of the complexes. From this, it can be speculated that the addition of methyl groups on the phenanthroline would intensify the bathochromic effect.

As chirality also contributes to the potent activity of the studied complexes, CD spectroscopy was utilized to confirm that the chirality of the ancillary ligand, *S*,*S*-DACH had been preserved after oxidation. According to the CD measurements obtained for **PCLB** (Appendix A), **5CLB** (Appendix A) and **56CLB** (Appendix A), it appears that the conjugation process did not induce an observable level of stereomutation. Negative absorption bands were mostly observed at lower wavelengths for the complexes, especially for **PCLB** (Appendix A). In addition, the CD spectra of **5CLB** (Appendix A) and **56CLB** (Appendix A) exhibited weak positive absorption bands at higher wavelengths. Overall, this pattern is consistent with previously published examples of platinum(IV) complexes that contain *S*,*S*-DACH. They also exhibit negative absorption bands at lower wavelengths and positive absorption bands at higher wavelengths [57,58,59,60,61,64]. In contrary, platinum(IV) complexes containing the enantiomer of *S*,*S*-DACH generally demonstrate positive absorption bands at lower wavelengths and negative absorption bands at higher wavelengths [60,64].

### 2.2. Stability

The stability of **PCLB**, **5CLB**, and **56CLB** was evaluated by HPLC. The complexes were tested in three solvent systems, 10 mM PBS (~7.4 pH), 0.9% NaCl and d.i.H_2_O/DMSO (50:50). This process was followed for 1 h at room temperature for each complex in each solvent system (Table 3 and Appendix A). No significant reduction was observed between 30 min and 1 h for the complexes per solvent system as shown in Table 3. However, it is noteworthy that when these complexes are left in solution for more extended periods, reduction to the corresponding platinum(II) species appears inevitable (Appendix A). This is likely influenced by the nature of the coordinated CLB ligand, which is highly susceptible to rapid hydrolysis due to its functional group [47,65].

### 2.3. Lipophilicity Studies

For lipophilicity measurements, HPLC was utilized instead of the traditional shake-flask method, as described [59,61]. A few studies have also reported the important and effective use of HPLC to measure the lipophilicity of compounds [66,67]. Each complex was eluted under several isocratic ratios. A standard curve was then generated to calculate the log value of the capacity factor (log k) according to eqn (1). Following eqn (2), the percentage of organic solvent (CH_3_CN) in the mobile phase from each isocratic run was plotted against log k_w_ (Appendix A). Since log k_w_ is the chromatographic lipophilicity index or measure of lipophilicity of the complex, a higher value corresponds to greater lipophilicity [59,68,69]. Since **PCLB**, **5CLB**, and **56CLB** are structurally related, there were no significant variations in lipophilicity noted based on the calculated log k_w_ values (Table 4). It is evident that **PCLB** is the least lipophilic of the group, with a calculated log k_w_ value of 1.80, which is likely due to the absence of methyl groups in its phenanthroline system. **5CLB** and **56CLB** were the most lipophilic of the group with calculated log k_w_ values of 1.90 and 1.87, respectively, and this is due to the presence of methyl substituents in their phenanthroline systems. Particularly, a 0.1 difference between the calculated log k_w_ values of **PCLB** and **56CLB** was demonstrated, and this difference is consistent with previously published examples of platinum(IV) derivatives [59]. Additionally, the chromatographic lipophilicity index of CLB was also measured using the abovementioned method, with a calculated log k_w_ of 1.88, which is within the range observed for **PCLB**, **5CLB**, and **56CLB**. However, CLB was approximately 1-fold more lipophilic than **PCLB**. CLB also demonstrated a similar log k_w_ value as **5CLB** with only a 0.01 difference. The order of increasing lipophilicity is **PCLB** > **5CLB** > CLB > **56CLB**. Overall, the data suggest that the addition of a platinum(IV) scaffold to CLB has no significant influence on overall lipophilicity.

### 2.4. Reduction Experiments

Intracellularly, a platinum(IV) complex is reduced into its platinum(II) congener and axial ligands in the presence of biological reducing agents such as AsA or glutathione [33,61,70,71,72,73]. This report followed the reduction in **PCLB**, **5CLB**, and **56CLB** using ^1^H-NMR and 1D-^195^Pt-NMR spectroscopy. The method used for the reduction experiments was adapted from the literature with minor modifications [61,74,75,76]. A total of 10 mM PBS (~7.4 pH) was selected as media to mimic physiological conditions, while 10 mM AsA was used as the biological reducing agent. Prior to the reduction experiments, a control was considered where each metal complex was dissolved with PBS in D_2_O. 1D-^195^Pt-NMR was employed within the regions of −2800 and 400 ppm (30 min per region) at 37 °C. Typically, our platinum(II) complexes resonate within the −2800 ppm region (Appendix A), while their platinum(IV) derivatives resonate around the 400 ppm region (Appendix A) [57,58,59,60,61,63]. A control was required to ensure that the PBS would not interfere in the reduction process of the complexes. From the preliminary 1D-^195^Pt-NMR spectra obtained (Figure 11), the state of all complexes was unchanged, and no sign of reduction was observed.

The reduction property of **PCLB** was monitored for 1 h by ^1^H-NMR spectroscopy at 37 °C (Figure 12). Upon treatment of AsA, the metal complex started reducing at 5 min as shown in Figure 12. Utilizing the resonances of the phenanthroline protons (H2 and H9; H4 and H7; H5 and H6; H3 and H8) from both platinum(II) and (IV) species as a qualitative measure of hydrolysis, and considering that the resonances of these phenanthroline protons from both platinum(II) and (IV) species were almost equivalent, it is estimated that 40–50% reduction occurs within 10 min. At 1 h, the metal complex was almost reduced to its platinum(II) congener, **PHEN*SS***. A notable observation from the ^1^H-NMR experiments is the movement of resonances, particularly the upfield shift of the phenanthroline protons and the downfield shift of the aromatic protons of CLB (Figure 12).

Subsequently, after the ^1^H-NMR experiments, 1D-^195^Pt-NMR was also carried out centered on the regions of −2800 and 400 ppm, at 37 °C. As shown in Figure 13, no platinum(IV) resonance was recorded in the −400 ppm region, while a platinum(II) resonance was recorded in the −2800 ppm region, confirming the reduction in **PCLB** and the presence of the platinum(II) complex, **PHEN*SS***. To further confirm the formation of **PHEN*SS***, an aliquot of the NMR sample containing the reduced **PCLB** prodrug was analyzed by ESI-MS (Appendix A). The results indicate that **PCLB** could be reduced inside tumor cells and that it could exert antitumor effects via platinum(II) congener and CLB; therefore, it is a potential dual-action prodrug.

Furthermore, the reduction properties of **5CLB** and **56CLB** were also investigated with the expectation that these would reduce to their platinum(II) congeners slower than **PCLB** because of their phenanthroline. Theoretically, the methyl groups are expected to contribute to better stability and may aid in slowing the reduction in the complexes. Accordingly, the ^1^H-NMR spectra of **5CLB** and **56CLB** exhibited the same pattern as **PCLB**, wherein the movement of resonances, such as the upfield shift from the phenanthroline protons and the downfield shift from the aromatic protons of CLB were demonstrated (Appendix A). It was found that **5CLB** and **56CLB** reached 50% reduction at approximately 5 min (Appendix A), which is marginally faster compared to **PCLB**. This goes against the expectation that the methyl groups in the phenanthroline ligand would stabilize the complexes in the presence of a reducing agent such as AsA, hence decreasing the rate of reduction. Nonetheless, the 1D-^195^Pt-NMR spectra of **5CLB** (Appendix A) and **56CLB** (Appendix A) obtained after 1 h from the sequential ^1^H-NMR proton experiments, still corroborate that these complexes can be reduced inside tumor cells to their platinum(II) counterparts, **5ME*SS*** and **56ME*SS***, respectively, along with the CLB ligand to promote antitumor effects. Additionally, aliquots of the NMR samples containing both the reduced **5CLB** and **56CLB** prodrugs were also analyzed by ESI-MS to further confirm the formation of their corresponding platinum(II) species (Appendix A).

Theoretically, an effective prodrug should be able to withstand reduction prior to entering the cell or a specific target to demonstrate a maximal therapeutic effect. Having prodrugs that rapidly reduce would only defy the notion of inertness conferred by the platinum(IV) oxidation state, which has always been the selling point of designing platinum(IV) prodrugs. Overall, it is also important to note that these reduction experiments are only approximate and do not necessarily reflect how prodrugs such as **PCLB**, **5CLB**, and **56CLB** would behave in biological environments such as whole blood and blood serum.

### 2.5. Growth Inhibition Assays

**PCLB**, **5CLB**, and **56CLB**, together with their platinum(II) and (IV) precursors, cisplatin, oxaliplatin, carboplatin, and CLB, were screened for antiproliferative activity against a panel of cell lines, which included HT29 colon, U87 glioblastoma, MCF-7 breast, A2780 ovarian, H460 lung, A431 skin, Du145 prostate, BE2-C neuroblastoma, SJ-G2 glioblastoma, MIA pancreas, ADDP ovarian (cisplatin-resistant A2780 clone) and the non-tumor-derived MCF10A breast line. Growth inhibition was assessed using the MTT assay after 72 h of treatment. A summary of the determined GI_50_ values is reported in Table 5. As expected, **56ME*SS*** exhibited the greatest potency across all the screened cell lines, with a determined average GI_50_ value of 40 ± 0.01 nM, followed by **5ME*SS*** with GI_50_ values ranging from 30 to 0.32 nM. In comparison, **PHEN*SS*** also exhibited potency in the cell lines with an average GI_50_ value of 434 ± 0.11 nM. This correlates with previous reports hence once again indicating that methylated phenanthrolines of these atypical platinum(II) complexes contribute to the biological activity [39,56,77]. Furthermore, **56ME*SS***, **5ME*SS***, and **PHEN*SS*** were significantly more potent than cisplatin, oxaliplatin, carboplatin, and CLB in the entire cell line panel. **56ME*SS*** was more than 2000-fold more active than cisplatin in the ADDP ovarian cell line, while **5ME*SS*** and **PHEN*SS*** were 800-fold and 150-fold more active, respectively, in the same cell line. Relative to CLB, the biological activity of **56ME*SS***, **5ME*SS***, and **PHEN*SS*** are far greater. For instance, **56ME*SS*** was estimated to be 4000-fold more active than CLB in the HT29 colon and MIA pancreas cell lines. In the Du145 prostate cell line, **56ME*SS*** displayed a GI_50_ value of 4.6 ± 0.0004 nM, which is a 6000-fold increase in potency compared to CLB (27,000 ± 3 nM). It is noteworthy that CLB has a weaker biological activity in all cell lines tested compared to cisplatin and oxaliplatin, with GI_50_ values ranging from 4300 to 40,000 nM. For example, CLB has a determined GI_50_ value of 40,000 ± 2 nM in the MIA pancreas cell line where cisplatin was 5-fold more active (7500 ± 2 nM) and oxaliplatin was 44-fold more active (900 ± 0.2 nM). While CLB was potent in the A2780 ovarian cell line (4300 ± 0.4 µM), cisplatin and oxaliplatin were still 3–4 times more active than CLB. Nevertheless, it should be noted that CLB is significantly better than carboplatin, as reflected in the results wherein carboplatin displayed GI_50_ values above 50,000 nM in six cell lines (HT29 colon, U87 glioblastoma, MCF-7 breast, MIA pancreas, ADDP cisplatin-resistant ovarian variant, and MCF10A normal breast cell lines).

The platinum(IV) dihydroxy complexes, **PHEN*SS*(IV)(OH)_2_**, **5ME*SS*(IV)(OH)_2_**, and **56ME*SS*(IV)(OH)_2_** outperformed cisplatin, oxaliplatin, carboplatin, and CLB over the all cell lines tested, with average GI_50_ values of 3300 ± 0.88 nM, 443 ± 0.08 nM and 151 ± 0.05 nM, respectively. Evidently, these platinum(IV) dihydroxy complexes are less cytotoxic than their platinum(II) congeners as the OH molecules in the axial positions of these complexes do not have any inherent biological activity, yet they still demonstrate potency, which is due to improved stability while the reducing properties are selectively better in the hypoxic environments of cancer cells [56]. The platinum(IV) derivatives coordinated to the DNA-alkylating agent CLB, **PCLB**, **5CLB**, and **56CLB** displayed remarkable GI_50_ values, especially in the case of **56CLB**, where its determined average GI_50_ value of 40 ± 0.01 nM was the same as **56ME*SS*** in all cell lines. **56CLB** elicited its strongest biological activity in seven cell lines representative of HT29 colon (6 ± 0.002 nM), Du145 prostrate (3 ± 0.001 nM), MCF10A normal breast (7 ± 0.004 nM), MIA pancreas (11 ± 0.003 nM), H460 lung (13 ± 0.006 nM), A2780 ovarian (21 ± 0.005 nM) and the cisplatin-resistant ovarian variant, ADDP (10 ± 0.003 nM) (Table 5). In addition, **56CLB** was determined to be 1–4 times more potent than its potent platinum(II) congener, **56ME*SS***, in these seven cell lines. Notably, **56CLB** was at least 440-fold more potent than cisplatin, and 10,000-fold more potent than CLB in the Du145 prostate cell line, which is a substantial difference. **PCLB** and **5CLB** however are less potent than **56CLB**, but still demonstrated low GI_50_ values ranging from 14 to 680 nM in the entire cell line tested. Nonetheless, **PCLB** and **5CLB** still proved to be unparalleled in terms of cytotoxicity compared to the prodrugs designed by Qin et al. and Ma et al., specifically in the MCF-7 cell line [51,52]. Particularly, **56CLB** (63 ± 0.015 nM) proved to be about 63-fold and 200-fold more potent than the cisplatin-CLB (3990 ± 0.40 nM) and oxaliplatin-CLB (12,650 ± 0.98 nM) prodrugs designed by Qin et al., respectively [52]. In the same cell line, the cisplatin prodrug incorporating two CLB ligands (2190 ± 0.02 nM) designed by Ma et al. [51] was 35-fold less potent than **56CLB**, 12-fold less potent than **5CLB** (180 ± 0.017 nM) and 3-fold less potent than **PCLB**. Overall, the GI_50_ values obtained for **PCLB**, **5CLB**, and **56CLB** across all the human cancer cell lines tested in this study justify that our heterocyclic, non-DNA coordinating platinum(II) complexes (**PHEN*SS***, **5ME*SS***, and **56ME*SS***) provide better vehicular scaffolding than cisplatin or oxaliplatin and most importantly, generate prodrugs with superior biological activity. Moreover, these prodrugs are equally sensitive in the cisplatin-resistant ADDP cell line in comparison to all other cell line populations, indicating that the prodrugs are not susceptible to the drug resistance mechanisms induced by this standard clinical treatment.

### 2.6. ROS Production

A subtle balance of intracellular ROS levels is critical for cancer cell function [78,79]. The significant production of ROS above basal level compromises cell function by inducing damage to DNA and perturbing the DNA damage response, ensuing mitochondrial outer membrane permeabilization [80,81,82,83]. Cellular stress induces the cleavage of the DNA repair protein system, Poly (ADP-ribose) polymerase (full length 116 kDa) by executioner caspases (3, 6, and 7) into 89 and 24 kDa subunits. These two fragments generate a reduced DNA-binding affinity and repairing ability, consequently promoting apoptosis [83,84]. Additionally, ROS can also stimulate DNA damage by nucleoside oxidation, which primes base transversions, degrades mitochondrial DNA by prompting lesions or strand breaks, and activate the intrinsic apoptotic pathway to induce cell death [84]. Here we report the ROS potential of **PCLB**, **5CLB**, and **56CLB**, together with their platinum(II) and (IV) scaffolds, as well as cisplatin and CLB in the human cancer cell line, HT29 colon. The HT29 colon cell line was selected as it is one of the human cancer cell lines wherein **PCLB**, **5CLB**, and **56CLB** demonstrated superb antitumor potential (Table 5).

HT29 colon cells were stained by DCFH-DA and then treated with the compounds at each specific GI_50_ concentration. Upon ROS production, the DCFH-DA produced dichlorodihydrofluorescein, a fluorescent product. The measured fluorescence was, therefore, proportional to the produced ROS. HT29 colon cells treated with precursor platinum(II) complexes, **PHEN*SS***, **5ME*SS***, and **56ME*SS*** showed an increase in ROS at 24, 48, and 72 h (Appendix A), with a significant increase in those treated with **5MESS** and **56ME*SS*** at 24 and 72 h (Appendix A). Additionally, cells treated with precursor platinum(IV) dihydroxy complexes, **PHEN*SS*(IV)(OH)_2_**, **5ME*SS*(IV)(OH)_2_**, and **56ME*SS*(IV)(OH)_2_** also showed a significant increase in ROS at 24, 48 and 72 h (Table 6 and Appendix A). Furthermore, cells treated with the platinum(IV)-CLB complexes, **PCLB**, **5CLB**, and **56CLB** showed a remarkable increase in ROS by 24 h and continued to increase up to 72 h (Figure 14 and Appendix A) when compared to the control. Peak ROS production by TBHP was observed at 24 h (514 RFU), while a decrease in ROS was observed at 48 h (336 RFU) and 72 h (332 RFU) while still being significant in comparison to control (Table 6, Figure 14 and Appendix A). Control cells showed an RFU of ~50. **56CLB** at 24 h reported 395 RFU, 437 RFU at 48 h, and 558 RFU at 72 h (Table 6,
Figure 14 and Appendix A), respectively, showing a significant production of ROS in comparison to control cells. **5CLB** at 24 h produced 337 RFU, 324 RFU at 48 h, and 376 RFU at 72 h (Table 6, Figure 14 and Appendix A), respectively. ROS production by **PCLB** was 190 RFU at 24 h, but significantly increased at 48 h (244 RFU) and 72 h (285 RFU) (Table 6, Figure 14 and Appendix A). CLB showed no significant ROS production until 48 h with 190 RFU in comparison to control cells. Overall, these results confirm that the platinum(IV)-CLB complexes **PCLB**, **5CLB**, and **56CLB** are expected to induce DNA damage through the production of ROS inside cancer cells.

## 3. Materials and Methods

All chemicals and reagents were of spectroscopic grade and used without further purification. The deionized water (d.i.H_2_O) used for the experiments was obtained from a MilliQ^TM^ system (Millipore Australia Pty Ltd, Sydney, NSW, Australia). Potassium tetrachloroplatinate(II) (K_2_PtCl_4_), *S*,*S-*DACH, 1,10-phenanthroline, 5-methyl-1,10-phenanthroline, 5,6-dimethyl-1,10-phenanthroline, CLB, *N*,*N′*-dicyclohexylcarbodiimide (DCC), dimethyl sulfoxide (DMSO), acetonitrile (CH_3_CN), trifluoroacetic acid (TFA), silver nitrate (AgNO_3_^−^), ascorbic acid (AsA) and 30% hydrogen peroxide (H_2_O_2_) were purchased from Sigma-Aldrich, Sydney, NSW, Australia. Sep-Pak^®^ C_18_-reverse phase columns were purchased from Waters Australia Pty Ltd, Sydney, NSW, Australia. Deuterated solvents such as deuterium oxide 99.9% (D_2_O) and deuterated acetonitrile (CD_3_CN) were purchased from Cambridge Isotope Laboratories, Andover, MA, USA. Methanol (MeOH) was obtained from Honeywell Research Chemicals, NJ, USA. Ethanol (EtOH), diethyl ether (Et_2_O), ethyl acetate (EtOAc) and acetone were purchased from ChemSupply, Gillman, SA, Australia. All other chemicals, reagents, and lab consumables were purchased from commercial sources.

### 3.1. Instrumentation

#### 3.1.1. Flash Chromatography

A Biotage Isolera^TM^ One flash chromatography system equipped with a Biotage^®^ Sfär C18 D (Duo 100 Å 30 μm 30 g) was utilized to purify the platinum(IV) complexes. The mobile phase consisted of solvents, A (d.i.H_2_O) and B (MeOH). The samples were dissolved in d.i.H_2_O/MeOH (50:50) and eluted through the column with a 0–30% linear gradient for 50 min with a flow rate of 4 mL/min, collected within the set wavelengths of 200–400 nm.

#### 3.1.2. High-Performance Liquid Chromatography (HPLC)

An Agilent Technologies 1260 Infinity instrument equipped with a Phenomenex Onyx^TM^ Monolithic C_18_-reverse phase column (100 × 4.6 mm, 5 µm pore size) or an Agilent ZORBAX RX-C_18_ column (100 × 4.6 mm, 3.5 µm pore size) was utilized. The mobile phase consisted of solvents, A (0.06% TFA in d.i.H_2_O) and B (0.06% TFA in CH_3_CN/d.i.H_2_O (90:10)). An injection volume of 5 µL was utilized and eluted with a 0–100% linear gradient over 15 min with a flow rate of 1 mL/min, at the set wavelengths of 214 and 254 nm.

The stability of **PCLB**, **5CLB**, and **56CLB** was monitored using an Agilent Technologies 1260 Infinity instrument equipped with a Phenomenex Onyx^TM^ Monolithic C_18_-reverse phase column (100 × 4.6 mm, 5 µm pore size). The mobile phase consisted of solvents, A (0.06% TFA in d.i.H_2_O) and B (0.06% TFA in CH_3_CN/d.i.H_2_O (90:10)). An injection volume of 5 µL was utilized and eluted with a 0–100% linear gradient over 15 min with a flow rate of 1 mL/min, at the set wavelengths of 214 and 254 nm. Each complex was dissolved separately in three solvent systems including, 10 mM phosphate-buffered saline (PBS) at ~7.4 pH, 0.9% sodium chloride (NaCl) or standard saline solution and d.i.H_2_O/DMSO (50:50). Each HPLC run per solvent system was followed for 1 h at room temperature.

For lipophilicity measurements, analytical HPLC was utilized. Elution profiles were acquired on an Agilent Technologies 1260 Infinity instrument equipped with a Phenomenex Onyx^TM^ Monolithic C_18_-reverse phase column (100 × 4.6 mm, 5 µm pore size). The mobile phase consisted of solvents, A (0.06% TFA in d.i.H_2_O) and B (0.06% TFA in CH_3_CN/d.i.H_2_O (90:10)). Potassium iodide was used as an external dead volume marker to determine the dead time of the column. Retention times (T_R_) were measured at varying isocratic ratios ranging from 42% to 48% of solvent B at 1 mL/min. An injection volume of 10 µL was utilized. Capacity factors were determined according to Equation (1):(1)$$k=(T_{R}-T_{0})/T_{0},$$
where k is the capacity factor, T_R_ is the retention time of the analyte, and T_0_ represents the dead time. A minimum of four different mobile compositions were used for each complex to calculate k. A linear plot was generated of log k against the concentration of CH_3_CN in the mobile phase to determine the value of log k_w_ expressed by Equation (2):(2)$$\log k=S\varphi +\log k_{w},$$
where S is the slope, φ is the concentration of the CH_3_CN in the mobile phase and log k_w_ represents the capacity factor of the complex in 100% d.i.H_2_O. Extrapolation of this linear plot to the *y*-intercept indicates the log k_w_ value.

#### 3.1.3. Nuclear Magnetic Resonance (NMR) Spectroscopy

^1^H-NMR, 2D-COSY, ^1^H-^195^Pt-HMQC, and 1D-^195^Pt-NMR were carried out on a 400 MHz Bruker Avance spectrometer at 298 K. All complexes were prepared to a concentration of 10 mM in 600 µL using D_2_O. ^1^H-NMR was set to 10 ppm and 16 scans with a spectral width of 8250 Hz and 65,536 data points. 2D-COSY was acquired using a spectral width of 3443 Hz for both ^1^H nucleus, F1, and F2 dimensions, with 256 and 2048 data points, respectively. ^1^H-^195^Pt-HMQC was carried out using a spectral width of 214,436 Hz and 256 data points for ^195^Pt nucleus, F1 dimension, also a spectral width of 4808 Hz with 2048 data points for ^1^H nucleus, F2 dimension. 1D-^195^Pt was measured using a spectral width of 85,470 Hz and 674 data points. All recorded resonances were presented as chemical shifts in parts per million (δ ppm) with *J*-coupling constants reported in Hz. For spin multiplicity: s (singlet); d (doublet); dd (doublet of doublets); t (triplet); q (quartet); qd (quartet of doublets) and m (multiplet). All spectroscopic data gathered were generated and plotted using Bruker TopSpin 4.1.3 analysis software.

The reduction behavior of **PCLB**, **5CLB**, and **56CLB** was monitored using ^1^H-NMR and 1D-^195^Pt-NMR spectroscopy. A sequence of ^1^H-NMR experiments was carried out for 1 h at 37 °C, followed by 1D-^195^Pt-NMR within the regions of −2800 and 400 ppm. An amount of 10 mM PBS (~7.4 pH) was transferred to a vial and reduced to dryness through rotary evaporation. AsA (~1 mg) was combined with the metal complex (~5 mg) and transferred to the vial containing the dried PBS. A total of 600 µL of D_2_O was then added to the vial to dissolve the complex, AsA and the PBS together. Each reaction was followed at 37 °C until the complete reduction in the prodrugs.

#### 3.1.4. Ultraviolet-Visible (UV) Spectroscopy

An Agilent Technologies Cary 3500 UV-Vis Multicell Peltier spectrophotometer was utilized to perform the UV spectroscopy experiments. UV spectra were recorded at room temperature in the range of 200–400 nm with a 1 cm quartz cuvette. All complexes were prepared in d.i.H_2_O. For the titration experiments, a stock solution of each complex (1 mM) was prepared and total aliquots of 9 × 3 µL were titrated into a cuvette containing d.i.H_2_O (3000 µL). Experiments were repeated in triplicate. All spectra were baseline corrected and average extinction coefficients (ε) were determined with standard deviation and errors based on the generated plot curves.

#### 3.1.5. Circular Dichroism (CD) Spectroscopy

A Jasco J-810 CD spectropolarimeter was used to measure the CD spectra of the studied prodrugs. The samples were prepared in d.i.H_2_O in a 1 mm optical glass cuvette or 1 cm quartz cuvette. CD experiments were undertaken at room temperature in the wavelength range of 200–400 nm (20 accumulations) with a bandwidth of 1 nm, data pitch of 0.5 nm, a response time of 1 s and a 100 nm/min scan speed. The flowrate of nitrogen gas was 6 L/min. A CD simulation tool (CDToolX) was used to generate the spectra.

#### 3.1.6. Electrospray Ionization Mass Spectrometry (ESI-MS)

ESI-MS experiments were undertaken using a Waters SYNAPT G2-Si quadruple time-of-flight (QTOF) HDMS. A stock solution of each complex (1 mM) was prepared in d.i.H_2_O and 5 µL of the solution was diluted with 995 µL of d.i.H_2_O to create the sample solution. The wire or capillary, where the sample solutions were injected, was washed with d.i.H_2_O/CH_3_CN (50:50) before every experiment to avoid cross-contamination.

### 3.2. Chemistry

#### 3.2.1. General Synthesis of CLB Anhydride

The anhydride of CLB was synthesized as reported [54], with minor adjustments. CLB (401 mg; 1.20 mmol) was reacted with 1 mol eq. of DCC (275 mg; 1.20 mmol) in acetone (30 mL) for 72 h at room temperature. Vacuum filtration was performed to remove the dicyclohexylurea (DCU) by-product, leaving a colorless filtrate that was rotary evaporated to afford a sticky precipitate. The precipitate was diluted with EtOAc (20 mL) and left to settle at 4 °C for 48 h to remove excess DCU. The final solution was filtered using a syringe and concentrated into a thick transparent oil through rotary evaporation. Yield: 699 mg; 89.7%. HPLC, T_R_: 17 min.

#### 3.2.2. General Synthesis of Precursor Platinum(II) Complexes of Type, [Pt^II^(H_L_)(A_L_)]Cl_2_

**[Pt^II^(H_L_)(A_L_)]^2+^** was synthesized accordingly [57,58,59,60,61,62], with minor adjustments. K_2_PtCl_4_ (900 mg; 2.20 mmol) and 1 mol eq. of *S*,*S*-DACH (250 mg; 2.20 mmol) were reacted in d.i.H_2_O (50 mL), and sonicated until completely dissolved, to produce a clear dark orange solution. The solution was left to settle at 4 °C for 48 h. The resulting yellow precipitate was collected by vacuum filtration and washed with d.i.H_2_O (20 mL), followed by EtOH (30 mL) and subsequently Et_2_O (40 mL) to yield [Pt(*S*,*S*-DACH)]Cl_2_ (758 mg; 92%). [Pt(*S*,*S*-DACH)]Cl_2_ was used without further purification and it was refluxed with 1.1 mol eq. of H_L_ (1,10-phenanthroline, 5-methyl-1,10-phenanthroline or 5,6-dimethyl-1,10-phenanthroline) in d.i.H_2_O (200–250 mL) for 48 h at 100 °C. The resulting clear and dark yellow solution was filtered, and the volume was reduced to approximately 5 mL through rotary evaporation. Purification was achieved using a Waters (2g) C_18_-reverse-phase Sep-pak^®^ column, which was activated with MeOH (10 mL), and washed using d.i.H_2_O (20 mL) before eluting the concentrated metal complex solution. Elution was carried out using a Bio-Rad EM-1 Econo^TM^ pump equipped with UV monitor at a rate of 2.80 mL/min. It is important to note that the flow rate may change depending on the concentration of the metal complex solution. The isolated yellow band was collected and evaporated to dryness through rotary evaporation, affording the isolated products as fine yellow powders.

**PHEN*SS***–Yield: 625 mg; 96%. ^1^H-NMR (400 MHz, D_2_O_,_ δ): 8.88 (d, H2 and H9, 2H, *J =* 5.4 Hz), 8.85 (d, H4 and H7, 2H, *J* = 8.3 Hz), 8.09 (s, H5 and H6, 2H), 7.97 (dd, H3 and H8, 2H, *J*_1_ = 8.3 Hz, *J*_2_ = 5.4 Hz), 2.70 (m, H1’ and H2’, 2H), 2.20 (d, H3’ and H6’ eq., 2H), 1.64 (d, H4’ and H5’ eq., 2H), 1.46 (d, H3’ and H6’ ax., 2H), and 1.23 (m, H4’ and H5’ ax., 2H). ^1^H-^195^Pt-HMQC (400 MHz, D_2_O_,_ δ): 8.88/−2821 ppm; 8.09 ppm/−2821 ppm.

**5ME*SS***–Yield: 751 mg; 94%. ^1^H-NMR (400 MHz, D_2_O_,_ δ): 8.80 (m, H2 and H9; H4, 3H), 8.60 (d, H7, 1H, *J* = 8.3 Hz), 7.95 (m, H3 and H8, 2H), 7.59 (s, H6, 1H), 2.71 (m, H1’ and H2’, 2H), 2.65 (s, CH_3_, 3H), 2.22 (d, H3’ and H6’ eq., 2H), 1.65 (d, H4’ and H5’ eq., 2H), 1.49 (d, H3’ and H6’ ax., 2H), and 1.24 (m, H4’ and H5’ ax., 2H). ^1^H-^195^Pt-HMQC (400 MHz, D_2_O_,_ δ): 8.80/−2810 ppm; 7.95/−2810 ppm.

**56ME*SS***–Yield: 536 mg; 92%. ^1^H-NMR (400 MHz, D_2_O_,_ δ): 8.87 (d, H2 and H9, 2H, *J =* 8.5 Hz), 8.78 (d, H4 and H7, 2H, *J* = 5.2 Hz), 7.95 (dd, H3 and H8, 2H, *J*_1_ = 8.6 Hz, *J*_2_ = 5.4 Hz), 2.60 (m, H1’ and H2’, 2H), 2.59 (s, 2 × CH_3_, 6H), 2.22 (d, H3’ and H6’ eq., 2H), 1.66 (d, H4’ and H5’ eq., 2H), 1.48 (d, H3’ and H6’ ax., 2H), and 1.03 (m, H4’ and H5’ ax., 2H). ^1^H-^195^Pt-HMQC (400 MHz, D_2_O_,_ δ): 8.87/−2822 ppm.

#### 3.2.3. General Synthesis of Precursor Platinum(IV) Dihydroxy Complexes of Type, [Pt^IV^(H_L_)(A_L_)(OH)_2_](NO_3_)_2_

**[Pt^IV^(H_L_)(A_L_)(OH)_2_]^2+^** was synthesized accordingly [57,58,59,60,61], with minor adjustments. **[Pt^II^(H_L_)(A_L_)]Cl_2_** was treated with 2 mol eq. of AgNO_3_^−^ in d.i.H_2_O (100 mL). The reaction solution was stirred continuously for one week at room temperature in the dark. The white precipitate by-product, silver chloride (AgCl) was filtered either through syringe filtration or vacuum filtration to afford a clear yellow filtrate containing **[Pt^II^(H_L_)(A_L_)](NO_3_)_2_**. The filtrate was treated with 30% H_2_O_2_ (10 mL) and refluxed for 3 h at 70 °C in the dark. The reaction solution was lyophilized and left overnight to afford a pale yellow to brown precipitate. A small amount of d.i.H_2_O (3–5 mL) was added to the reaction precipitate, which was briefly agitated with ultrasonic waves, followed by the addition of excess CH_3_CN to afford the final product, **[Pt^IV^(H_L_)(A_L_)(OH)_2_](NO_3_)_2_**, which appears as a fine white powder when dried. To save time and potentially minimize product loss upon collection, vacuum filtration should be avoided and alternatively, employ centrifugation instead.

**PHEN*SS*(IV)(OH)_2_**–Yield: 470 mg; 93%. ^1^H-NMR (400 MHz, D_2_O_,_ δ): 9.23 (d, H2 and H9, 2H, *J =* 6.2 Hz), 9.09 (d, H4 and H7, 2H, *J* = 9 Hz), 8.34 (s, H5 and H6, 2H), 8.27 (dd, H3 and H8, 2H, *J*_1_
*=* 8.3 Hz, *J*_2_ = 5.5 Hz), 3.18 (m, H1’ and H2’, 2H), 2.39 (d, H3’ and H6’ eq., 2H), 1.69 (m, H4’ and H5’ eq.; H3’ and H6’ ax., 4H), and 1.32 (m, H4’ and H5’ ax., 2H). ^1^H-^195^Pt-HMQC (400 MHz, D_2_O_,_ δ): 9.23/439 ppm; 8.27/439 ppm.

**5ME*SS*(IV)(OH)_2_**–Yield: 657 mg; 91%. ^1^H-NMR (400 MHz, D_2_O_,_ δ): 9.20 (d, H2, 1H, *J* = 5.5 Hz), 9.17 (d, H4, 1H, *J* = 8.5 Hz), 9.11 (d, H9, 1H, *J* = 5.4 Hz), 8.95 (d, H7, 2H, *J* = 8.3 Hz), 8.28 (q, H3, 1H), 8.20 (q, H8, 1H), 8.15 (s, H6, 1H), 3.16 (m, H1’ and H2’, 2H), 2.90 (s, CH_3_, 3H), 2.38 (d, H3’ and H6’ eq., 2H), 1.67 (m, H4’ and H5’ eq.; H3’ and H6’ ax., 4H), and 1.30 (m, H4’ and H5’ ax., 2H). ^1^H-^195^Pt-HMQC (400 MHz, D_2_O_,_ δ): 9.20/430 ppm; 9.11/430 ppm; 8.28/430 ppm; 8.20/430 ppm.

**56ME*SS*(IV)(OH)_2_**–Yield: 752 mg; 90%. ^1^H-NMR (400 MHz, D_2_O_,_ δ): 9.20 (d, H2 and H9, 2H, *J =* 8.6 Hz), 9.11 (d, H4 and H7, 2H, *J* = 5.5 Hz), 8.23 (dd, H3 and H8, 2H, *J*_1_ = 8.6 Hz, *J*_2_ = 5.5 Hz), 3.16 (m, H1’ and H2’, 2H), 2.85 (s, 2 × CH_3_, 6H), 2.38 (d, H3’ and H6’ eq., 2H), 1.68 (m, H4’ and H5’ eq.; H3’ and H6’ ax., 4H), and 1.32 (m, H4’ and H5’ ax., 2H). ^1^H-^195^Pt-HMQC (400 MHz, D_2_O, δ): 9.20/422 ppm; 8.23/422 ppm.

#### 3.2.4. Synthesis of Platinum(IV) Complexes of Type, [Pt^IV^(H_L_)(A_L_)(CLB)(OH)](NO_3_)_2_

**[Pt^IV^(H_L_)(A_L_)(OH)_2_](NO_3_)_2_** was reacted with 2 mol eq. of CLB anhydride in DMSO (1–2 mL) for 72 h at room temperature in the dark. The reaction solution was washed with excess Et_2_O and vigorously mixed using a plastic pipette, followed by centrifugation to afford a clear supernatant and an oily brown layer. The supernatant was discarded while the oily brown layer was dissolved in methanol (1–2 mL), followed by the addition of excess Et_2_O to induce precipitation. Centrifugation was undertaken to collect the final precipitate that appeared pasty and dark red-orange in color. Excess acetone was mixed with the precipitate and sonicated, affording a pure and solidified red-orange precipitate that was collected through centrifugation.

**PCLB**–Yield: 225 mg; 92%. ^1^H-NMR (400 MHz, D_2_O_,_ δ): 9.32 (d, H2, 1H, *J =* 5.5 Hz), 9.27 (d, H9, 1H, *J =* 5.5 Hz), 8.99 (dd, H4 and H7, 2H, *J*_1_
*=* 8.3 Hz, *J*_2_ = 3.3 Hz), 8.24 (m, H3 and H8, 2H), 8.20 (s, H5 and H6, 2H), 6.37 (q, a and b; c and d, 4H, *J =* 8.8 Hz), 3.51 (m, e, f, g, and h, 8H), 3.17 (m, H1’ and H2’, 2H), 2.38 (d, H3’ and H6’ eq., 2H), 1.97 (t, α, 2H), 1.68 (m, H4’ and H5’ eq.; H3’ and H6’ ax.; γ, 6H), and 1.27 (m, H4’ and H5’ ax.; β, 4H). ^1^H-^195^Pt-HMQC (400 MHz, D_2_O_,_ δ): 9.30/542 ppm; 8.24/542 ppm. HPLC, T_R_: 9.13 min. UV λ_max_ nm (ε/M·cm^−1^ ± SD × 10^4^, d.i.H_2_O): 279 (2.92 ± 0.82), 203 (9.48 ± 2.20). CD λ_max_ nm (Δε/M·cm^−1^ × 10^1^, d.i.H_2_O): 206 (−404), 258 (−2.03), 277 (−68.5). ESI-MS: *calculated* for [M-H]^+^: *m*/*z* = 807.2; *experimental*: *m*/*z* = 808.2.

**5CLB**–Yield: 194 mg; 82%. ^1^H-NMR (400 MHz, D_2_O_,_ δ): 9.26 (qd, H2 and H9, 2H), 9.05 (dd, H4, 1H, *J*_1_
*=* 8.4 Hz, *J*_2_ = 4.3 Hz), 8.87 (dd, H7, 1H, *J*_1_
*=* 8.3 Hz, *J*_2_ = 3.6 Hz), 8.26 (m, H3, 1H), 8.19 (m, H8, 1H), 7.95 (d, H6, 1H, *J* = 4.5 Hz), 6.27 (q, a and b; c and d, 4H, *J* = 8.3 Hz), 3.48 (s, e, f, g, and h, 8H), 3.16 (m, H1’ and H2’, 2H), 2.72 (s, CH_3_, 3H), 2.38 (d, H3’ and H6’ eq., 2H), 1.97 (t, α, 2H), 1.46 (m, γ, 2H), 1.68 (m, H4’ and H5’ eq.; H3’ and H6’ ax., 4H), and 1.25 (m, H4’ and H5’ ax.; β, 4H). ^1^H-^195^Pt-HMQC (400 MHz, D_2_O_,_ δ): 9.26/544 ppm. HPLC, T_R_: 9.26 min. UV λ_max_ nm (ε/M.cm^−1^ ± SD × 10^4^, d.i.H_2_O): 284 (2.31 ± 1.48), 204 (6.99 ± 1.33). CD λ_max_ nm (Δε/M.cm^−1^ × 10^1^, d.i.H_2_O): 208 (−233), 232 (−7.77), 260 (+70), 286 (+2.42). ESI-MS: *calculated* for [M-H]^+^: *m*/*z* = 821.2; *experimental*: *m*/*z* = 822.2.

**56CLB**–Yield: 166 mg; 87%. ^1^H-NMR (400 MHz, D_2_O_,_ δ): 9.25 (d, H2, 1H, *J =* 5.4 Hz), 9.22 (d, H9, 1H, *J =* 5.4 Hz), 9.10 (dd, H4 and H7, 2H, *J*_1_
*=* 8.6 Hz, *J*_2_ = 5 Hz), 8.23 (q, H3 and H8, 2H, *J* = 5.5 Hz), 6.24 (q, a and b; c and d, 4H, *J =* 8.6 Hz), 3.52 (s, e, f, g, and h, 8H), 3.17 (m, H1’ and H2’, 2H), 2.63 (d, 2 × CH_3_, 6H, *J =* 3.6 Hz), 2.39 (d, H3’ and H6’ eq., 2H), 1.97 (t, α, 2H), 1.69 (m, H4’ and H5’ eq.; H3’ and H6’ ax., 4H), and 1.20 (m, H4’ and H5’ ax.; β; γ, 6H). ^1^H-^195^Pt-HMQC (400 MHz, D_2_O_,_ δ): 9.25/532 ppm; 9.22/532 ppm; 8.23/532 ppm. HPLC, T_R_: 9.45 min. UV λ_max_ nm (ε/M.cm^−1^ ± SD × 10^4^, d.i.H_2_O): 291 (2.53 ± 1.23), 204 (8.25 ± 1.77). CD λ_max_ nm (Δε/M.cm^−1^ × 10^1^, d.i.H_2_O): 207 (−259), 234 (−138), 240 (−375), 260 (+581), 291 (−224). ESI-MS: *calculated* for [M-H]^+^: *m*/*z* = 835.2; *experimental*: *m*/*z* = 835.2.

### 3.3. Biological Investigations

#### 3.3.1. Cell Growth Assays

Cell growth assays were performed at the Calvary Mater Newcastle Hospital, NSW, Australia. The cell lines tested were: HT29 colon, U87 glioblastoma, MCF-7 breast, A2780 ovarian, H460 lung, A431 skin, Du145 prostate, BE2-C neuroblastoma, SJ-G2 glioblastoma, MIA pancreas, ADDP ovarian (cisplatin-resistant A2780 clone) and the non-tumor-derived MCF10A breast line. In addition to **PCLB**, **5CLB**, and **56CLB**, all precursor platinum(II) and (IV) complexes, cisplatin, oxaliplatin, carboplatin, as well as CLB were tested for reference. All test agents were prepared in DMSO (30 mM stock solutions) and stored at −20 °C until use. All cell lines were cultured in a humidified atmosphere 5% carbon dioxide at 37 °C. The cancer cell lines were maintained in Dulbecco’s modified Eagle’s medium (DMEM) (Trace Biosciences, Australia) supplemented with 10% fetal bovine serum, 10 mM sodium bicarbonate penicillin (100 IU mL^−1^), streptomycin (100 µg mL^−1^) and glutamine (4 mM). The non-cancer MCF10A cell line was cultured in DMEM:F12 (1:1) cell culture media, 5% heat inactivated horse serum, supplemented with penicillin (50 IU mL^−1^), streptomycin (50 µg mL^−1^), 20 mM 4-(2-hydroxyethyl)-1-piperazineethanesulfonic acid (HEPES), L-glutamine (2 mM), epidermal growth factor (20 ng mL^−1^), hydrocortisone (500 ng mL^−1^), cholera toxin (100 ng mL^−1^) and insulin (10 μg mL^−1^). Cytotoxicity was determined by plating cells in duplicate in 100 mL medium at a density of 2500–4000 cells per well in 96-well plates. On day 0 (24 h after plating) when the cells were in logarithmic growth, 100 μL of medium with or without the test agent was added to each well. After 72 h, the GI_50_ was evaluated using the MTT (3-[4,5-dimethylthiazol-2-yl]-2,5-diphenyltetrazolium bromide) assay and absorbance was read at 540 nm. An eight-point dose response curve was produced from which the GI_50_ value was calculated, representing the drug concentration at which cell growth was inhibited by 50% based on the difference between the optical density values on day 0 and those at the end of drug exposure [85].

#### 3.3.2. Reactive Oxygen Species (ROS) Detection Assay

To investigate for the presence of ROS in treated cells, DCFDA/H2DCFDA-cellular ROS Assay Kit (Abcam, Cambridge, MA, USA) was used, as described [86,87,88]. A total of 25,000 cells/mL of HT29 cells in DMEM were seeded in 96-well plates. Cells were washed with 1X kit buffer, then stained with 25 μM 2’,7’-dichlorofluorescein diacetate (DCFH-DA) and incubated for 45 min. DCFH-DA was then removed, and cells were then re-washed with 1X kit buffer, after which phenol red free media was added. Cells were then treated with GI_50_ drug concentration for each complex. The plates were directly scanned to measure fluorescence (relative fluorescence units (RFU)) at different time points using the Glo-Max^®^-Multimode microplate reader (Promega Corporation, Alexandra, VIC, Australia) at an excitation/emission of 485/535 nm. To generate the positive control (20 μM tert-butyl hydroperoxide (TBHP)), cells were washed with 1X kit buffer, and stained with DCFDA (25 μM) for 45 min. DCFDA was then removed and TBHP was added in phenol red free media. The plate was scanned to measure the RFU at different time points as mentioned above. The same method applies for the negative control treated with 20 μM *N*-acetylcysteine (NAC).

## 4. Conclusions

The structure and purity of non-symmetrical platinum(IV) derivatives containing the DNA alkylator, CLB, were confirmed by HPLC, NMR, UV, CD, and ESI-MS. The use of NMR provided insight into the reduction property of the platinum(IV)-CLB complexes. HPLC was utilized for stability experiments and lipophilicity measurements. The order of increasing lipophilicity was **PCLB** > **5CLB** > **56CLB**. A ROS detection assay was also undertaken to demonstrate the production of ROS of the complexes in the HT29 colon human cancer cell line. **56CLB** exhibited the highest ROS production. According to the in vitro results, the studied complexes exhibited superior activity in the cell lines tested. Remarkably, **56CLB** demonstrated the highest activity in HT29 colon, Du145 prostate, MCF10A normal breast, MIA pancreas, H460 lung, A2780, and ADDP ovarian cell lines with GI_50_ values ranging between 2.7 and 21 nM. A correlation was also drawn from the lipophilicity, in vitro, and ROS results. As **56CLB** was the most active platinum(IV) derivative, it exhibited the highest chromatographic lipophilicity index and the greatest ROS production. In contrast, **PCLB** was the least active platinum(IV) derivative and demonstrated the lowest chromatographic lipophilicity index and the weakest ROS production. These observations are attributed to the structural variances of the complexes, mainly the presence or absence of methyl groups of phenanthroline. In vivo evaluation and cellular uptake studies of these complexes have been scheduled for the near future.

## 5. Patents

This work is part of Australian Provisional Patent Application 2022900110, Platinum(IV) complexes, February 2022, Western Sydney University, Sydney, Australia.

## Figures and Tables

**Figure 1 ijms-23-10471-f001:**
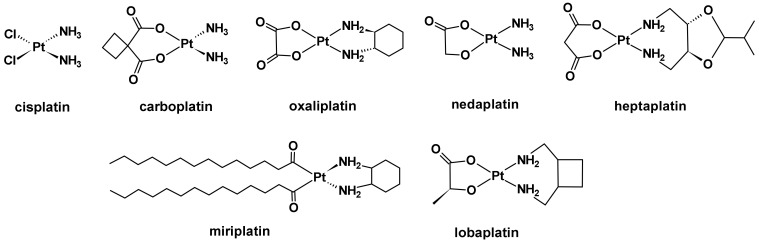
Chemical structures of platinum(II) chemotherapeutics.

**Figure 2 ijms-23-10471-f002:**
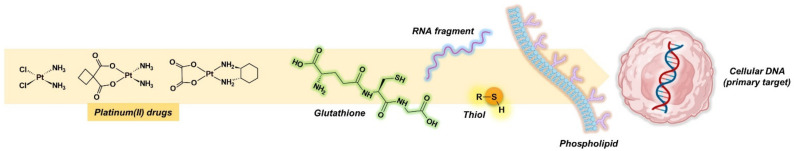
An illustration of the clinically used platinum(II) drugs and the likely molecular interactions before it reaches the primary target, the DNA.

**Figure 3 ijms-23-10471-f003:**
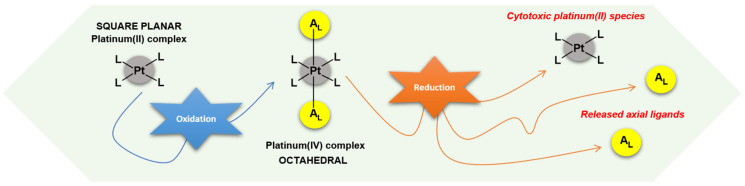
Platinum(II) complexes undergo oxidation to generate their platinum(IV) derivatives, subsequently platinum(IV) complexes undergo reduction to generate their platinum(II) congeners, as well as release the bioactive or non-bioactive axial ligands. L = ligand. A_L_ = axial ligand.

**Figure 4 ijms-23-10471-f004:**
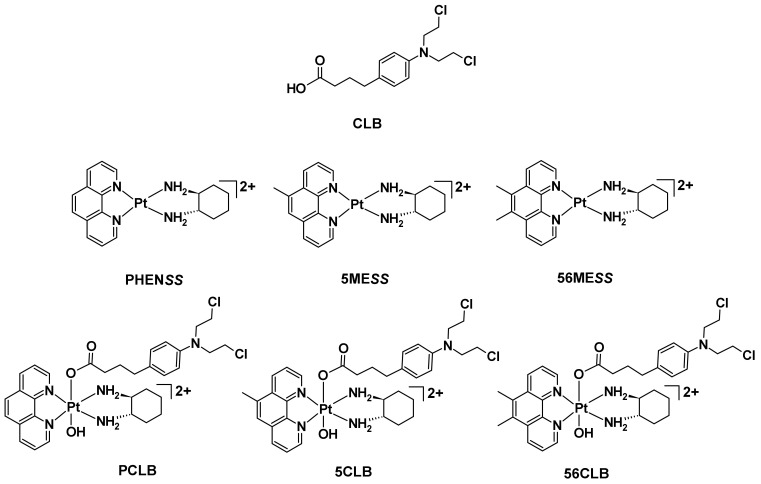
Structures of CLB, unconventional non-covalent DNA-binding platinum(II) complexes, **PHEN*SS***, **5ME*SS***, and **56ME*SS***, together with the studied platinum(IV)-CLB prodrugs, **PCLB**, **5CLB**, and **56CLB**.

**Figure 5 ijms-23-10471-f005:**
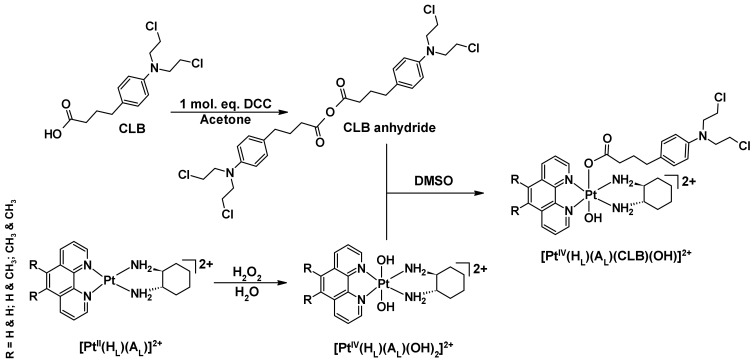
Synthetic pathway to create the studied platinum(IV)-CLB prodrugs. Counter-ions of the platinum complexes are omitted for clarity.

**Figure 6 ijms-23-10471-f006:**
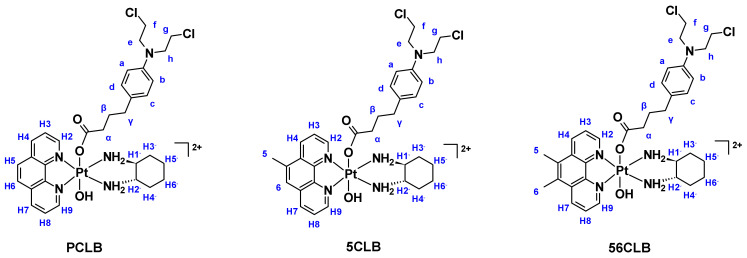
Proton labeling system of the platinum(IV)-CLB complexes.

**Figure 7 ijms-23-10471-f007:**
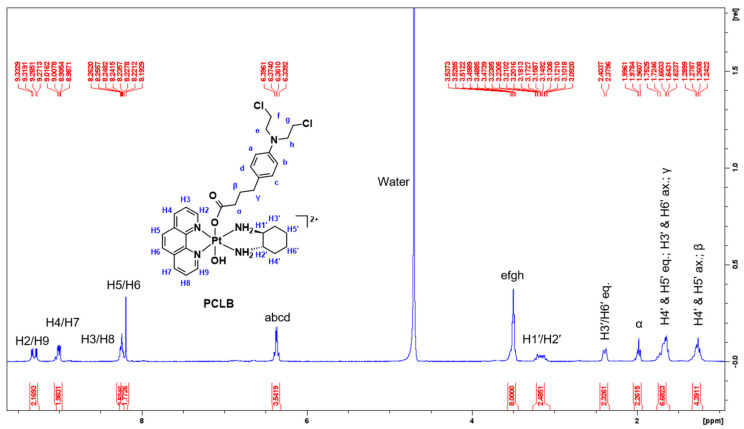
^1^H-NMR spectrum of **PCLB** in D_2_O obtained at 298 K, with proton assignment. Inset: structure of **PCLB** with proton labeling system.

**Figure 8 ijms-23-10471-f008:**
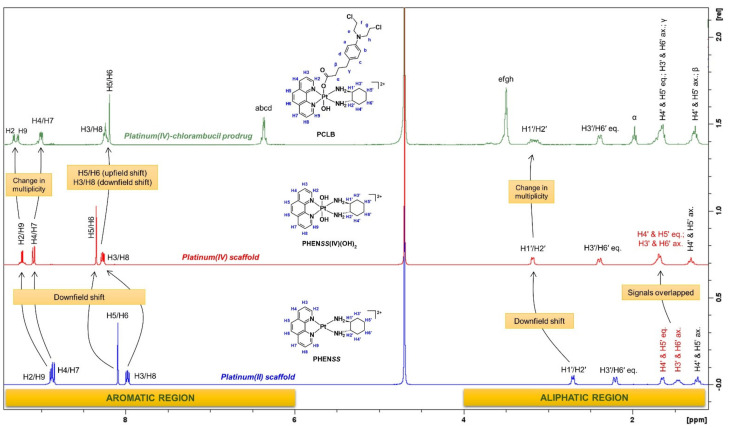
Stacked ^1^H-NMR spectra of **PHEN*SS***, **PHEN*SS*(IV)(OH)_2_** and **PCLB** in D_2_O obtained at 298 K, with arrows highlighting the movement of chemical resonances and change in multiplicity. Inset: structures of **PHEN*SS***, **PHEN*SS*(IV)(OH)_2_** and **PCLB** with proton labeling systems.

**Figure 9 ijms-23-10471-f009:**
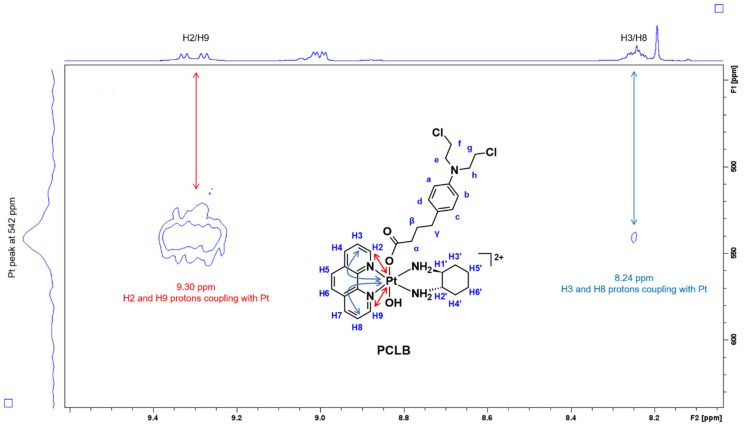
Expanded ^1^H-^195^Pt-HMQC spectrum of **PCLB**, highlighting the coupling between H2, H9, H3, and H8 protons with platinum. Region: 400 ppm. Inset: structure of **PCLB** with proton labeling system and arrows that indicate coupling.

**Figure 10 ijms-23-10471-f010:**
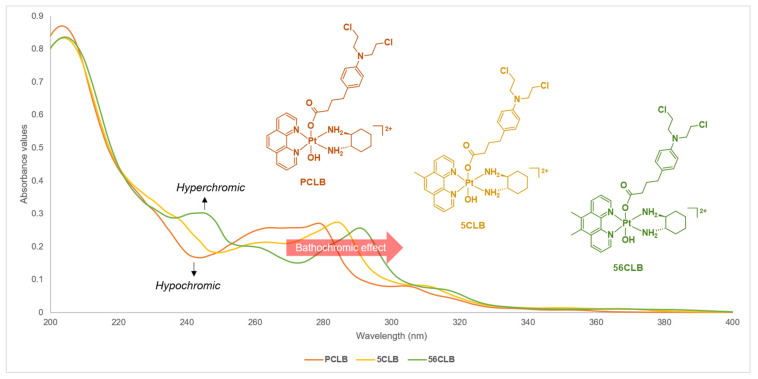
Stacked UV spectra of **PCLB**, **5CLB**, and **56CLB** obtained at 298 K, showing UV absorptions at different wavelengths. Inset: color-coded structures of **PCLB**, **5CLB**, and **56CLB**.

**Figure 11 ijms-23-10471-f011:**
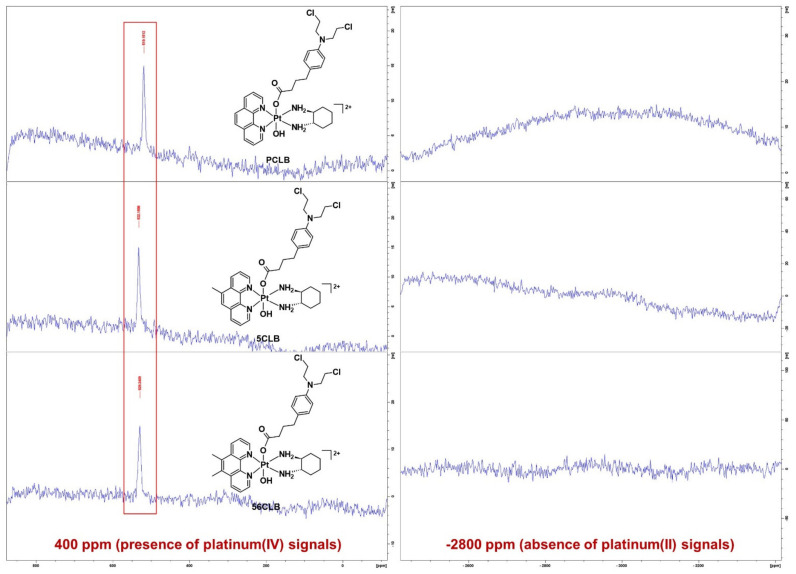
1D-^195^Pt-NMR spectra of **PCLB**, **5CLB**, and **56CLB** in 10 mM PBS (~7.4 pH) within the regions of −2800 and 400 ppm at 37 °C. The red box encloses the platinum resonances recorded for the complexes within the region of 400 ppm. Inset: structures of **PCLB**, **5CLB**, and **56CLB**.

**Figure 12 ijms-23-10471-f012:**
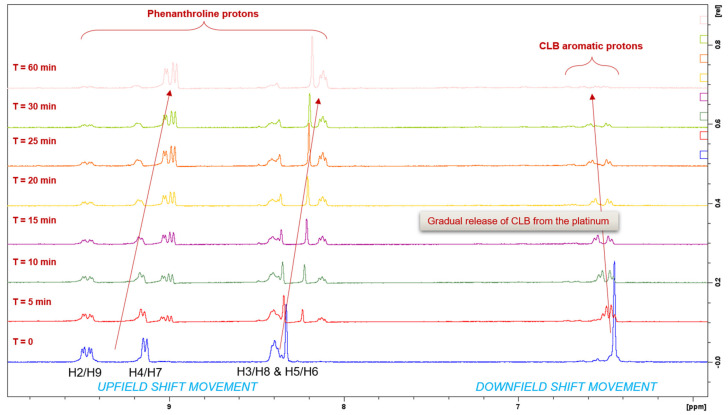
Zoomed ^1^H-NMR spectra of **PCLB** with PBS and AsA in D_2_O at 310.15 K, in different time intervals, highlighting the movement of resonances of the phenanthroline protons and the aromatic protons of the CLB ligand as indicated by the red arrows. **T** represents time in min.

**Figure 13 ijms-23-10471-f013:**
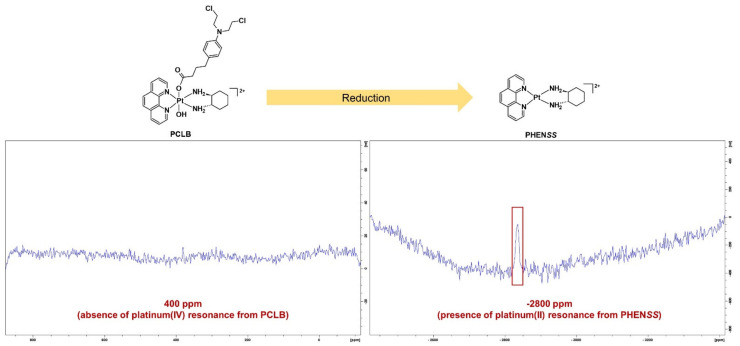
1D-^195^Pt-NMR spectra of **PCLB** with PBS and AsA in D_2_O at 310.15 K, within the regions of 400 and −2800 ppm, highlighting the complete reduction in the complex after 1 h. Inset: structures of **PCLB** and its platinum(II) scaffold, **PHEN*SS***.

**Figure 14 ijms-23-10471-f014:**
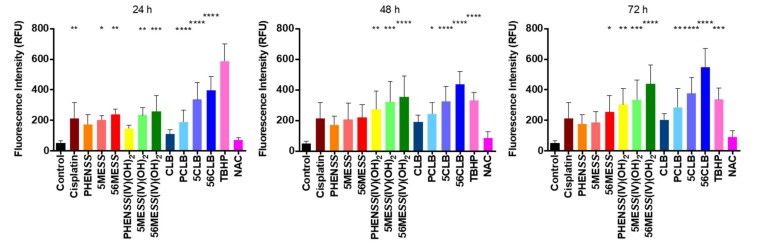
ROS production upon treatment with platinum(IV)-CLB prodrugs, platinum(II) and (IV) scaffolds, CLB, and cisplatin in HT29 colon cells at 24, 48, and 72 h. **** indicates *p* < 0.0001 compared with control. *** indicates *p* < 0.001 compared with control. ** indicates *p* < 0.01, compared with control. * indicates *p* < 0.05 compared with control. Data points denote mean ± SEM. n = 3 from three independent experiments where samples were run in triplicate.

**Table 1 ijms-23-10471-t001:** Summary of the ^1^H-NMR and ^1^H-^195^Pt-HMQC data of platinum(IV)-CLB complexes including chemical shifts (δ ppm), multiplicity, integration and *J*-coupling constants (Hz). Since D_2_O was used in the experiments, no amine resonances were observed due to proton exchange.

Proton Labels	PCLB	5CLB	56CLB
**H2**	9.32 (d, 1H, *J =* 5.5 Hz)	9.26 (qd, 2H)	9.25 (d, 1H, *J =* 5.4 Hz)
**H9**	9.27 (d, 1H, *J =* 5.5 Hz)		9.22 (d, 1H, *J =* 5.4 Hz)
**H4**	8.99 (dd, 2H, *J*_1_ *=* 8.3, *J*_2_ = 3.3 Hz)	9.05 (dd, 1H, *J*_1_ *=* 8.4, *J*_2_ = 4.3 Hz)	9.10 (dd, 2H, *J*_1_ *=* 8.6, *J*_2_ = 5 Hz)
**H7**		8.87 (dd, 1H, *J*_1_ *=* 8.3, *J*_2_ = 3.6 Hz)	
**H3**	8.24 (m, 2H)	8.26 (m, 1H)	8.23 (q, 2H, *J =* 5.5 Hz)
**H8**		8.19 (m, 1H)	
**H5 and H6**	8.20 (s, 2H)	-	-
**H6**	-	7.95 (d, 1H, *J* = 4.5 Hz)	-
**a and b**	6.37 (q, 4H, *J =* 8.8 Hz) (resonances overlapped)	6.27 (q, 4H, *J* = 8.3 Hz) (slight distance between a and b and c and d resonances)	6.24 (q, 4H, *J =* 8.6 Hz) (greater distance between a and b and c and d resonances)
**c and d**			
**e, f, g, and h**	3.51 (m, 8H) (sharp resonance)	3.48 (s, 8H)	3.52 (s, 8H)
**H1’ and H2’**	3.17 (m, 2H)	3.16 (m, 2H)	3.17 (m, 2H)
**CH_3_** **(5 positions)**	-	2.72 (s, 3H)	-
**CH_3_** **(5 and 6 positions)**	-	-	2.63 (d, 6H, *J =* 3.6 Hz)
**H3’ and H6’ eq.**	2.38 (d, 2H)	2.38 (d, 2H)	2.39 (d, 2H)
**α**	1.97 (t, 2H)	1.97 (t, 2H)	1.97 (t, 2H)
**β**	Overlapping with H4’ and H5’ ax.	Overlapping with H4’ and H5’ ax.	Overlapping with H4’ and H5’ ax.
**γ**	Overlapping with H4’ and H5’ eq.; H3’ and H6’ ax.	1.46 (m, 2H)	Overlapping with H4’ and H5’ ax.
**H4’ and H5’ eq.; H3’ and H6’ ax.**	Overlapping with γ	1.68 (m, 4H)	-
**H4’ and H5’ ax.**	Overlapping with β	Overlapping with β	-
**H4’ and H5’ eq.; H3’ and H6’ ax.**	-	1.68 (m, 4H)	1.69 (m, 4H)
**H4’ and H5’ eq.; H3’ and H6’ ax.; γ**	1.68 (m, 6H)	-	-
**H4’ and H5’ ax.; β**	1.27 (m, 4H)	1.25 (m, 4H)	-
**H4’ and H5’ ax.; β; γ**	-	-	1.20 (m, 6H)
**^1^H/^195^Pt**	9.30, 8.24/542 ppm	9.26/544 ppm	9.25, 9.22, 8.23/532 ppm

**Table 2 ijms-23-10471-t002:** A summary of the absorbance shifts (nm) observed in the UV and CD spectra of the platinum(IV)-CLB complexes.

Platinum(IV)-CLB Complexes	UV λ_max_ nm (ε/M.cm^−1^ ± SD × 10^4^)	CD λ_max_ nm (Δε/M.cm^−1^ × 10^1^)
**PCLB**	279 (2.92 ± 0.82), 203 (9.48 ± 2.20)	206 (−404), 258 (−2.03), 277 (−68.5)
**5CLB**	284 (2.31 ± 1.48), 204 (6.99 ± 1.33)	208 (−233), 232 (−7.77), 260 (+70), 286 (+2.42)
**56CLB**	291 (2.53 ± 1.23), 204 (8.25 ± 1.77)	207 (−259), 234 (−138), 240 (−375), 260 (+581), 291 (−224)

**Table 3 ijms-23-10471-t003:** A summary of HPLC peak areas of the studied platinum(IV)-CLB complexes at 254 nm in PBS (~7.4 pH), 0.9% NaCl and d.i.H_2_O/DMSO at 30 min and 1 h, at room temperature.

Platinum(IV)-CLB Complexes	Solvent Systems, Time Points, and HPLC Peak Areas at 254 nm					
	PBS (~7.4 pH)		0.9% NaCl		d.i.H_2_O/DMSO	
	30 min	1 h	30 min	1 h	30 min	1 h
**PCLB**	85.0%	83.1%	89.4%	87.5%	93.2%	92.4%
**5CLB**	68.8%	67.3%	73.9%	72.4%	76.5%	75.8%
**56CLB**	78.2%	75.2%	84.6%	82.5%	85.0%	84.3%

**Table 4 ijms-23-10471-t004:** A summary of log k_w_ values of the studied platinum(IV)-CLB complexes and CLB.

Compounds	log k_w_
**PCLB**	1.80
**5CLB**	1.87
**56CLB**	1.90
CLB	1.88

**Table 5 ijms-23-10471-t005:** A summary of the GI_50_ values (nM) of the platinum(IV)-CLB complexes, together with precursor platinum(II) and (IV) scaffolds, CLB, cisplatin, oxaliplatin, and carboplatin in multiple cell lines. nd = not determined.

	GI_50_ Values (nM)												
**Platinum(IV)-CLB Prodrugs**	**HT29**	**U87**	**MCF-7**	**A2780**	**H460**	**A431**	**Du145**	**BE2-C**	**SJ-G2**	**MIA**	**MCF10A**	**ADDP**	**Average GI_50_ Values**
**PCLB**	170 ± 0.046	1200 ± 0.28	680 ± 0.043	300 ± 0.048	460 ± 0.063	510 ± 0.10	210 ± 0.12	440 ± 0.030	320 ± 0.040	190 ± 0.032	300 ± 0.045	290 ± 0.037	423 ± 0.07
**5CLB**	19 ± 0.0054	250 ± 0.015	180 ± 0.017	46 ± 0.015	39 ± 0.013	90 ± 0.015	14 ± 0.004	230 ± 0.039	180 ± 0.02	34 ± 0.005	22 ± 0.0045	26 ± 0.0064	94 ± 0.01
**56CLB**	6.1 ± 0.0020	93 ± 0.0067	63 ± 0.015	21 ± 0.0047	13 ± 0.0056	34 ± 0.0037	2.7 ± 0.0012	140 ± 0.012	130 ± 0.03	11 ± 0.003	7 ± 0.004	10 ± 0.0028	44 ± 0.01
**Platinum(II) complexes**													
**PHEN*SS***	160 ± 0.045	980 ± 0.27	1500 ± 0.50	230 ± 0.030	360 ± 0.035	480 ± 0.17	100 ± 0.038	380 ± 0.046	330 ± 0.066	200 ± 0.057	300 ± 0.058	190 ± 0.047	434 ± 0.11
**5ME*SS***	33 ± 0.0038	320 ± 0.026	200 ± 0.012	61 ± 0.010	41 ± 0.005	120 ± 0.025	22 ± 0.0027	270 ± 0.038	220 ± 0.01	48 ± 0.002	30 ± 0.0018	34 ± 0.0023	117 ± 0.01
**56ME*SS***	10 ± 0.0016	35 ± 0.0064	93 ± 0.044	76 ± 0.057	21 ± 0.0019	29 ± 0.0010	4.6 ± 0.00039	59 ± 0.0041	66 ± 0.022	13 ± 0.0020	16 ± 0.0012	13 ± 0.0022	36 ± 0.01
**Platinum(IV) dihydroxy complexes**													
**PHEN*SS*(IV)(OH)_2_**	710 ± 0.30	4900 ± 0.61	16,000 ± 4.5	800 ± 0.084	1700 ± 0.20	4300 ± 0.53	310 ± 0.092	3000 ± 0.53	1700 ± 0.35	3400 ± 2.2	1700 ± 0.20	1300 ± 0.35	3318 ± 0.88
**5ME*SS*(IV)(OH)_2_**	60 ± 0.0055	900 ± 0.058	1200 ± 0.39	240 ± 0.009	60 ± 0.005	360 ± 0.058	41 ± 0.0049	1400 ± 0.3	640 ± 0.07	160 ± 0.029	130 ± 0.019	130 ± 0.022	443 ± 0.08
**56ME*SS*(IV)(OH)_2_**	36 ± 0.0071	190 ± 0.023	480 ± 0.14	59 ± 0.0071	190 ± 0.15	120 ± 0.022	15 ± 0.0026	240 ± 0.022	210 ± 0.045	43 ± 0.0025	61 ± 0.0073	170 ± 0.12	151 ± 0.05
**DNA-targeting chemotherapeutics**													
Cisplatin	11,300 ± 1.9	3800 ± 1.1	6500 ± 0.8	1000 ± 0.1	900 ± 0.2	2400 ± 0.3	1200 ± 0.1	1900 ± 0.2	400 ± 0.1	7500 ± 1.3	5200 ± 0.52	28,000 ± 1.6	5842 ± 0.61
Oxaliplatin	900 ± 0.2	1800 ± 0.2	500 ± 0.1	160 ± 0.1	1600 ± 0.1	4100 ± 0.5	2900 ± 0.4	900 ± 0.2	3000 ± 1.2	900 ± 0.2	nd	800 ± 0.1	1463 ± 0.32
Carboplatin	>50,000	>50,000	>50,000	9200 ± 2.9	14,000 ± 1.0	24,000 ± 2.2	15,000 ± 1.2	19,000 ± 1.2	5700 ± 0.2	>50,000	>50,000	>50,000	32,242 ± 1.45
CLB	39,000 ± 2	38,000 ± 2	11,000 ± 1.9	4300 ± 0.4	11,000 ± 2	35,000 ± 0	27,000 ± 3	20,000 ± 1	11,000 ± 2	40,000 ± 2	17,000 ± 1	16,000 ± 1	22,000 ± 1.52

**Table 6 ijms-23-10471-t006:** ROS production upon treatment with platinum(IV)-CLB prodrugs, platinum(II) and (IV) scaffolds, CLB, TBHP, NAC, cisplatin, and control in HT29 colon cells at 24, 48, and 72 h.

Compounds	ROS Production in Different Time Intervals (RFU)		
	24 h	48 h	72 h
Control	50 ± 2	52 ± 4	57 ± 5
**PHEN*SS***	174 ± 2	172 ± 9	176 ± 7
**5ME*SS***	204 ± 4	205 ± 3	188 ± 3
**56ME*SS***	240 ± 5	218 ± 3	255 ± 4
**PHEN*SS*(IV)(OH)_2_**	144 ± 5	273 ± 4	303 ± 1
**5ME*SS*(IV)(OH)_2_**	234 ± 1	323 ± 9	335 ± 2
**56ME*SS*(IV)(OH)_2_**	259 ± 3	356 ± 11	438 ± 7
CLB	110 ± 3	190 ± 5	204 ± 4
**PCLB**	190 ± 2	244 ± 7	285 ± 3
**5CLB**	337 ± 8	324 ± 5	376 ± 1
**56CLB**	395 ± 3	437 ± 2	558 ± 2
Cisplatin	214 ± 9	214 ± 9	200 ± 9
TBHP	514 ± 3	336 ± 2	332 ± 5
NAC	71 ± 2	85 ± 6	91 ± 5

## Data Availability

All data relevant to the publication are included.

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
