# Peer review of "Potent Chlorambucil-Platinum(IV) Prodrugs"

_ijms, 2022, doi:10.3390/ijms231810471_

Round 1
Reviewer 1 Report
1) It would be better to include more information about the discovery of transition metal complex-based drugs in the introduction. For example, a) Nat. Chem., 11 (2019) 1041-1048ï¼› b) Chem Sci., 11 (2020) 11404-11412; c) Nat. Commun., 11 (2020) 3262; d) J Am. Chem. Soc., 143 (2021) 20224-20240.
2) The chemical structure of PCLB, 5CLB and 56CLB are quite similar. Please explain why 56CLB was exceptionally potent over the other two compounds.
3) For the characterization of the platinum complexes, the authors chose D2O for the recording of NMR spectra. How is the solubility of these complex in aqueous solution? The stability of these complexes in aqueous solution should be provided for reference.
4) Line 645-650: The author suggested that PCLB could be finally reduced to platinum complex PHENSS. More characterization data (i.e. HRMS result) of this newly formed compound is highly recommended to add as supporting information.
5) The authors investigated several properties of PCLB, 5CLB and 56CLB, such as lipohilicity and ROS result. In this case, which one of them played the key role in exerting their anticancer activity?
Author Response
Reviewer 1
- Comment 1: It would be better to include more information about the discovery of transition metal complex-based drugs in the introduction, a) Nat. Chem., 11 (2019) 1041-1048; b) Chem Sci., 11 (2020) 11404-11412; c) Nat. Commun., 11 (2020) 3262; d) J Am. Chem. Soc., 143 (2021) 20224-20240.
Author response: The reviewer provided a great suggestion, since information on this will allow readers and the authors to recognize the depth of coordination chemistry in cancer research. This is an abundant area of research and plenty of metal coordination complexes apart from platinum have been discovered and gained scientific interest due to their unique therapeutic properties. See Lines 39 – 45 for the new information we have added in the revised manuscript.
- Comment 2: The chemical structure of PCLB, 5CLB and 56CLB are quite similar. Please explain why 56CLB was exceptionally potent over the other two compounds.
Author response: The potency of the studied prodrugs, PCLB, 5CLB and 56CLB is directly related to the potency of their precursor platinum(II) complexes [(1,10-phenanthroline)(1S,2S-diaminocyclohexane) platinum(II)] dichloride (PtIIPHENSS), [(5-methyl-1,10-phenanthroline)(1S,2S-diaminocyclohexane)platinum(II)] dichloride (PtII5MESS) and [(5,6-dimethyl-1,10-phenanthroline)(1S,2S-diaminocyclohexane)platinum(II)] dichloride (PtII56MESS) which have been the subject of several publications (Dalton Trans. 2004, (8), 1145-1152; J. Inorg. Biochem. 2007, 101, (7), 1049-1058; Dalton Trans. 2007, (43), 5055-5064; ChemMedChem 2007, 2, (4), 488-495; J. Med. Chem. 2009, 52, (17), 5474-5484). It is clear to us that the chirality and type of diamine has a significant impact on GI50 values as does the choice of heterocyclic ligand. The structural differences between PtIIPHENSS, PtII5MESS and PtII56MESS are modest with changes in the pKa of the coordinated nitrogen atoms that could modulate the overall positive 2 charge. It is quite possible that PtII56MESS is perfectly sized for a binding site with the methyl substituents in the best orientation and that PtII5MESS and PtIIPHENSS are an increasingly less perfect fit. We have also investigated the number and position of the methyl substituents and change the position or increase the number and the potency diminishes (J. Inorg. Biochem. 2007, 101, (7), 1049-1058). Whilst at the molecular surface and conformational standpoint 1,10-phen 5,6-dialkylation induces negligible impact, as denoted by pKa values (i.e., 1,10-phen = 4.9 vs 5,6-dimethyl-1,10-phen = 5.6), alkylation significantly abrogates N-lone pair localisation. As previously investigated by Ramírez-Delgado et al. (J. Mex. Chem. Soc. 2015, 59 (4), 282-293) and others, pKa values are congruent with a variety of other descriptors, which include π acceptor potential, charge accepting properties (µ+) and electro-accepting power (ω+) (high π acceptor character = high capacity to accept charge). Hence via extrapolation of the values equated for Cu(II)-Phen complexes (table below), demethylation confers a significant modulatory effect on the aforementioned parameters.
Ligand |
Eo |
I |
A |
η |
μ– |
μ+ |
ω– |
ω+ |
pKa |
5,6-dimethyl-1,10-phen |
0.011 |
7.91 |
-0.18 |
3.86 |
-5.98 |
-2.12 |
4.63 |
0.58 |
5.6 |
1,10-phen |
0.145 |
8.26 |
-0.23 |
4.02 |
-6.25 |
-2.24 |
4.87 |
0.62 |
4.93 |
Ionisation energy (I), electron affinity (A), chemical hardness(η), chemical potential governing donation (μ-) and accepting (μ+) process, electrodonating (ω-) and electroaccepting (ω+) powers. J. Mex. Chem. Soc. 2015, 59 (4), 282-293.
- Comment 3: For the characterization of the platinum complexes, the authors chose D2O for the recording of NMR spectra. How is the solubility of these complex in aqueous solution? The stability of these complexes in aqueous solution should be provided for reference.
Author response: The reviewer did not specify which “platinum complexes”, either platinum(II) and (IV) precursor complexes or the studied platinum(IV) prodrugs, PCLB, 5CLB and 56CLB. Because the focus of this study is on the prodrugs, we have added a statement in Section 3.1 (Lines 446 – 448) of the revised manuscript addressing their solubility in water and aqueous solutions such as PBS (~7.4 pH) and 0.9% NaCl. Additionally, the prodrugs are highly soluble in D2O. 0.6 mL of D2O dissolves about 5 grams of the metal complex. This was used to prepare the prodrugs in the reduction studies undertaken (see Section 2.2.3, Lines 252 – 255). For the reviewer’s interest, all our platinum(II) and platinum(IV) precursor complexes are highly water soluble (Dalton Trans. 2004, (8), 1145-1152; Dalton Trans. 2019, 48, (46), 17228-17240). Furthermore, the stability of the studied prodrugs in aqueous solutions (PBS (~7.4 pH) and 0.9% NaCl) were also provided in the manuscript (see Section 3.2, Table 3 and Supplementary Materials, Figure S32). We also reported in manuscript that the prodrugs reduce back to their corresponding platinum(II) species in aqueous solutions if left for extended periods (Lines 579 – 581) and to further support this claim, we have added additional data in the Supplementary Materials (see Figure S33).
- Comment 4: Line 645-650: The author suggested that PCLB could be finally reduced to platinum complex PHENSS. More characterization data (i.e. HRMS result) of this newly formed compound is highly recommended to as supporting information.
Author response: We have provided new characterization data (high resolution ESI-MS spectra) as additional evidence to support our claim that the prodrugs did reduce to their corresponding platinum(II) congeners in the presence of ascorbic acid (see Supplementary Materials, Figures S35, S40 and S41).
- Comment 5: The authors investigated several properties of PCLB, 5CLB and 56CLB, such as lipophilicity and ROS result. In this case, which one of them played a key role in exerting their exerting their anticancer activity.
Author response: There is no indication from the experiments performed in the study that only one characteristic plays a key role. At present, as opposed to a single parameter eliciting the predominating influence, we suspect that there exists a co-interdependency relationship between the abovementioned parameters (see Authors’ response in Comment 2) and cytotoxicity. For example, we speculate that whilst the lower ionisation energy value of 5,6-dimethyl-1,10-phen may modulate enhanced cellular diffusion, reduced chemical hardness may also confer enhanced intracellular latency.

Reviewer 2 Report
The presented review describes the syntheses of three novel platinum(II) complexes possessing 1,10-phenathroline (or its methyl derivatives) and 1S,2S-diaminocyclohexane ligands. These complexes were oxidized to the corresponding dihydroxy platinum(IV) compounds which, in turn, were transformed to the final platinum(IV)-chlorambucil prodrugs.
All these synthesized compounds were extremely carefully characterized using various spectroscopic techniques. Also, their stability, lipophilicity and reduction properties were established.
The obtained prodrugs, together with their platinum(II) and (IV) precursors as well as reference drugs (cis-platin, oxaliplatin, carboplatin and CLB), were screened for antiproliferative activity against numerous human cancer cell lines. It was established that the platinum(IV)-chlorambucil prodrugs, as well as their platinum(II) precursors, exhibited remarkable antitumor properties for multiple human cancer cell lines compared to the currently used platinum drugs.
Moreover, the Authors indicated the relationship between the structure of these compounds and their activities (SAR) indicating how structural changes affect the biological activities. The paper could be interesting for researchers specializing in the medicinal chemistry.
The manuscript is well written: the syntheses are described briefly but correctly, all schemes and figures included in the text are proper and carefully drawn.
Taking these facts into consideration this reviewer recommends the article for publication in International Journal of Molecular Sciences if the Authors revise the manuscript according to the comments shown in the attached file.

Author Response
Reviewer 2
- The comments and suggestions provided by the reviewer are in the image below, as well as in the returned annotated manuscript.
Author response: The authors thank the reviewer for the effort and time in providing extensive and valuable feedback throughout the manuscript. We are also appreciative with the reviewer’s assurance in our research work, especially with our methodology and the results we obtained from it. We have revised the manuscript considering all the editorial comments and suggestions provided by the reviewer and made appropriate amendments. Please see the revised manuscript, with annotations (or comments) as reference for the reviewer and the editor.
